# Contributions of changes in climatology and perturbation and the resulting nonlinearity to regional climate change

Sachiho A. Adachi [1], Seiya Nishizawa [1], Ryuji Yoshida [1,2], Tsuyoshi Yamaura[1], Kazuto Ando [1], Hisashi Yashiro [1], Yoshiyuki Kajikawa[1,2] & Hirofumi Tomita[1]

Future changes in large-scale climatology and perturbation may have different impacts on regional climate change. It is important to understand the impacts of climatology and perturbation in terms of both thermodynamic and dynamic changes. Although many studies have investigated the influence of climatology changes on regional climate, the significance of perturbation changes is still debated. The nonlinear effect of these two changes is also unknown. We propose a systematic procedure that extracts the influences of three factors: changes in climatology, changes in perturbation and the resulting nonlinear effect. We then demonstrate the usefulness of the procedure, applying it to future changes in precipitation. All three factors have the same degree of influence, especially for extreme rainfall events. Thus, regional climate assessments should consider not only the climatology change but also the perturbation change and their nonlinearity. This procedure can advance interpretations of future regional climates.

[1] RIKEN Advanced Institute for Computational Science, 7-1-26 Minatojima-minami-machi, Chuo-ku, Kobe 650–0047, Japan. [2] Research Center for Urban Safety and Security, Kobe University, 1-1 Rokkodai, Nada-ku, Kobe 657–8501, Japan. Correspondence and requests for materials should be addressed to S.A.A. (email: sachiho.adachi@riken.jp)

Recently, studies analysing in situ observational data have reported an increase in heavy rainfall events[1,2]. Further increases due to global warming are a concern[3,4]. Heavy rainfall events often lead to serious human losses and social damage by causing landslides and flooded rivers. To minimise such risks and develop adaptation strategies, there is an increasing need for future regional climate projections.

Changes in the amount, frequency and intensity of regional precipitation are affected by thermodynamic and dynamic changes in the large-scale atmospheric conditions. Increasing greenhouse gases are known to raise the air temperature by modifying the atmospheric radiative balance[5]. The temperature warming enriches the moisture content in the atmosphere, which leads to an increase in precipitation[6]. This type of changes in the atmosphere is defined as thermodynamic changes. At the same time, the shift of the energy balance induces modifications in the atmospheric circulation patterns and the frequency, intensity and track of synoptic disturbances[7,8]. Such changes are defined as dynamic changes.

To understand the causes of regional precipitation change, the influences of large-scale thermodynamic and dynamic changes need to be understood. Emori and Brown[9] attempted to evaluate the influences of thermodynamic and dynamic changes in the atmosphere on global precipitation by using results from multiple general circulation models (GCMs). They reported that thermodynamic change, rather than dynamic change, plays a dominant role for both the mean and extreme precipitation in most areas, especially in the mid latitudes and high latitudes.

Some studies have addressed these influences from another perspective using a regional climate model (RCM)[10–13]. In these studies, a large-scale atmospheric state is decomposed into mean states and fluctuations from it, and the influences of the decomposed components on the regional climate were subsequently evaluated by constraining a boundary condition of the RCM. In this paper, the mean and fluctuation components of the large-scale atmosphere are called climatology and perturbation, respectively.

The evaluation separating the influences of the climatology and perturbation changes can be interpreted from the viewpoint of the thermodynamic and dynamic changes in the large-scale state as follows. Here, we assume that one component is changed, while the other remains unchanged. The changes in climatology

between two climates include both the thermodynamic and dynamic changes. The thermodynamic change due to the climatology change corresponds to the increase in the atmospheric moisture content associated with temperature warming, while the dynamic change corresponds to changes in the large-scale flow pattern caused by global circulation such as Hadley circulation and the position and strength of the westerly jets. The perturbation change also includes both the thermodynamic and dynamic changes. The dynamic change due to the perturbation change refers to changes in the frequency, intensity and track of disturbances such as tropical and extratropical cyclones, while the thermodynamic change refers to the changes in the temperature and humidity of the disturbances, accompanied by their changes in track.

The pseudo-warming method is one approach for evaluating the influence of changes in climatology using an RCM[10,11,14]. This method uses a boundary condition that modifies only the climatology. Although the changes in both the precipitation amount and intensity have been explained mainly by the changes in climatology thus far, there is a possibility that changes in the perturbation also significantly influence the precipitation, especially its intensity, in a limited region. This is because the changes in the track and frequency of cyclones directly affect the region through which they pass. Because the changes in perturbation are not considered in the pseudo-warming method, the degree of influence of the perturbation change on the regional climate projection has become a concern when using this method[15].

A few studies have evaluated the contribution of future changes in perturbation to the regional climate change[12,13]. However, the significance of the perturbation changes is still debated. One reason for the contradictory results is a lack of a standard evaluation method; therefore, a sophisticated procedure is needed for more precisely evaluating the individual contributions of changes in climatology and perturbation.

We propose a new procedure for evaluating these contributions to regional climate change. The proposed procedure requires an experimental set consisting of four experiments: two direct dynamical downscaling (DDS) experiments for the present and future climates and two experiments with boundary conditions exchanging either climatology or perturbation components between the two climates. The DDS experiments are conventional downscaling simulations, for which boundary conditions are

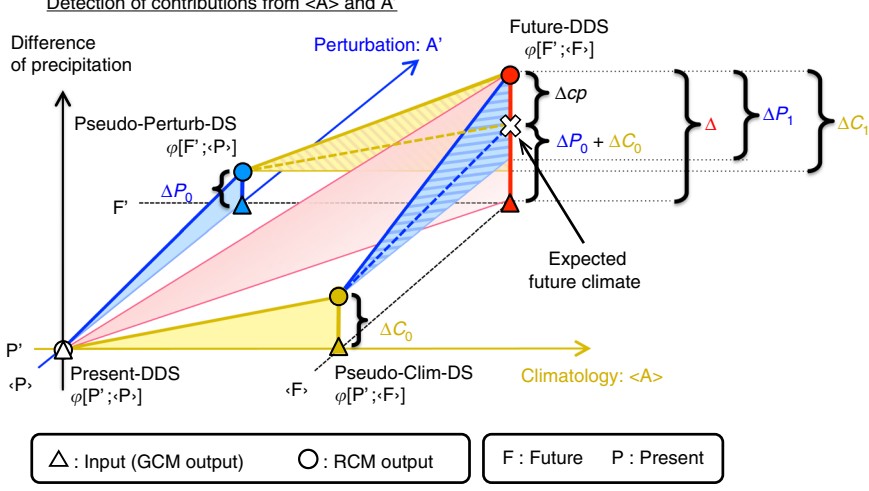

**Fig. 1** Diagram of impact assessment due to changes in climatology and perturbation. The triangles and circles indicate the boundary conditions and corresponding RCM outputs for each experiment, respectively. $\Delta$ is the future precipitation change estimated using the direct dynamical downscaling (DDS) method. $\Delta P$ and $\Delta C$ represent the contributions of changes in the perturbation and climatology, respectively. The expected climate change is defined by the sum of $\Delta P_0$ and $\Delta C_0$. $\Delta cp$ is the difference between the actual and expected future climates

---

**Table 1 Experimental design for estimating the contributions of climatology and perturbation changes and their nonlinear effect to regional climate change**

| Run name | Description of experiment | Boundary data |
|---|---|---|
| Present-DDS | Present climate experiment by DDS method | $P' + \langle P \rangle$ |
| Pseudo-Clim-DS | Pseudo-climatology-change downscaling experiment | $P' + \langle F \rangle$ |
| Pseudo-Perturb-DS | Pseudo-perturbation-change downscaling experiment | $F' + \langle P \rangle$ |
| Future-DDS | Future climate experiment by DDS method | $F' + \langle F \rangle$ |

$\langle\ \rangle$ and ' indicate the climatology and perturbation, respectively, of the large-scale boundary condition. P and F indicate the present and future climate data, respectively, provided by MRI-AGCM3.2S
*DDS* direct dynamical downscaling

---

given directly from GCM outputs. An analysis with four experiments allows not only for the influence of each component to be estimated but also for the influence due to the nonlinear effect between the two components to be extracted. Thus, by using this procedure, we can divide the causes of regional climate change into three factors: climatology change, perturbation change and the nonlinear effect between them. The demonstration of this procedure shows the importance of the symmetric treatment of the changes in climatology and perturbation to precisely understand the regional climate change. At the same time, the nonlinear effect also shows some significant influence, especially for extreme rainfall events.

## Results

**Principal concept and procedure.** Before presenting the procedure, we explain the principle, which was inspired by the theoretical basis provided in Nishizawa et al.[13]. Consider the decomposition of a large-scale atmospheric state A into the climatology and perturbation, $A = \langle A \rangle + A'$, where $\langle\ \rangle$ and ' denote the temporal average and fluctuation, respectively. Figure 1 shows a schematic diagram of the principal concept. The two horizontal axes represent the climatology and perturbation used as boundary conditions for an RCM. The vertical axis expresses the variable estimated by downscaling simulations, namely, the precipitation. The boundary condition and corresponding RCM output for each experiment are expressed by a triangle and circle, respectively.

Here, we aim at understanding the regional climate change between the present and future large-scale atmospheric states, P and F, obtained by a GCM. The RCM outputs using P and F as boundary conditions are denoted as functions $\varphi$ of each large-scale state: $\varphi[P'; \langle P \rangle]$ and $\varphi[F'; \langle F \rangle]$. These DDS simulations are named Present-DDS and Future-DDS for the present and future climate experiments, respectively, in this study.

As shown in Fig. 1, the total change in the regional climate between the present and future can be estimated as the difference between Present-DDS and Future-DDS:

$$\Delta = \varphi[F'; \langle F \rangle] - \varphi[P'; \langle P \rangle]. \quad (1)$$

We define the total change $\Delta$ as the "actual climate change". The total change includes the regional climate responses to large-scale atmospheric changes in both the climatology and perturbation and their nonlinear effect. Therefore, it is difficult to extract only the contribution of a change in a particular large-scale atmospheric component from the results of the two DDS experiments. To estimate the contribution of each component, two other boundary conditions need to be constructed: replacing only the climatology from P to F, i.e., $\langle F \rangle + P'$, and replacing only the perturbation, i.e., $\langle P \rangle + F'$. The RCM outputs constrained by these artificial boundary conditions can be expressed as $\varphi[P'; \langle F \rangle]$ and $\varphi[F'; \langle P \rangle]$. The experiment using the boundary condition of $\langle F \rangle + P'$ is designated the pseudo-climatology-change downscaling (Pseudo-Clim-DS) experiment, whereas that using $\langle P \rangle + F'$ is designated the pseudo-perturbation-change downscaling

(Pseudo-Perturb-DS) experiment. Table 1 summarises the above four simulations.

By comparing the four downscaling experiments, we can extract the individual contributions of the changes in climatology and perturbation from the regional climate responses. Based on the reference state P, the regional climate changes due to the large-scale climatology and perturbation changes are, respectively, described by

$$\Delta C_0 = \varphi[P'; \langle F \rangle] - \varphi[P'; \langle P \rangle], \quad (2)$$

$$\Delta P_0 = \varphi[F'; \langle P \rangle] - \varphi[P'; \langle P \rangle]. \quad (3)$$

Here, we define a regional climate change expressed only by a linear summation of $\Delta C_0$ and $\Delta P_0$ as the "expected climate change", corresponding to the cross in Fig. 1. The nonlinear effect between the climatology and perturbation changes is not considered in the expected climate change. If the difference between the actual climate change and expected climate change is denoted as $\Delta cp$, Eq. (1) can be expressed as follows:

$$\Delta = \Delta C_0 + \Delta P_0 + \Delta cp. \quad (4)$$

$\Delta cp$ in Eq. (4) has an important implication for the physical meaning. Regarding the two large triangles parallel to the climatology axis in Fig. 1, the height of the front triangle, $\Delta C_0$, is the regional climate response derived from Eq. (2). If the present perturbation is replaced by the future one under the same climatology changes, we can expect a different response, indicated as the height of the back triangle, $\Delta C_1$. The difference between the two estimations equals the nonlinear effect: $\Delta cp = \Delta C_1 - \Delta C_0$. The same is true with regard to the perturbation change; $\Delta P_0$ and $\Delta P_1$, respectively, are the differences by changing the perturbation under the present and future climatology and $\Delta cp = \Delta P_1 - \Delta P_0$. Previous studies[12,13] have attempted to compare the contributions of the climatology and perturbation changes by using $\Delta C_0$ and $\Delta P_1$. However, the nonlinear effect was not considered in these evaluations. Our procedure starts from nullifying the hypotheses of $\Delta C_1 = \Delta C_0$ and $\Delta P_1 = \Delta P_0$. The procedure considers the response to changes in perturbation as well as that to changes in climatology equivalently. This symmetric treatment of these two components reveals the nonlinear effect.

Next, we consider the implications of the three factors $\Delta C_0$, $\Delta P_0$ and $\Delta cp$ relative to the total change $\Delta$ with a focus on their signs. When $\Delta C_0 \cdot \Delta P_0 > 0$, these two contributions affect the regional climate in the same direction. When $\Delta C_0 \cdot \Delta P_0 < 0$, the sign of the expected climate change is determined by that of the regional climate change due to the component having a larger influence. The relationship between the signs of the nonlinear effect $\Delta cp$ and the expected climate change is important; compared to the expected climate change, the actual climate change is enhanced by $\Delta cp$ when $\Delta cp \cdot (\Delta C_0 + \Delta P_0) > 0$, whereas it is suppressed by $\Delta cp$ when $\Delta cp \cdot (\Delta C_0 + \Delta P_0) < 0$.

**Table 2 The precipitation indices evaluated in this study**

| Notation | Definition of Index |
|---|---|
| RAVE | 25-year mean of the daily precipitation |
| R1D | 25-year mean of the maximum 1-day precipitation |
| CDD | 25-year mean of the maximum number of consecutive dry days <br> (Note that a dry day is defined as one with a precipitation of <1 mm day$^{-1}$.) |

In short, the combination of the four downscaling experiments and the relationship described above allows the procedure to extract the contributions of the changes in the climatology and perturbation as well as the nonlinear effect to the regional climate change at a time.

**Demonstration of the proposed procedure.** To demonstrate the procedure, we applied it to future precipitation changes in western Japan. The Pacific side of western Japan has high rainfall in the summer season, which is characterised by typhoons, extratropical cyclones, seasonal stationary fronts and local rainfall due to convective instability. SCALE-RM[16–18], which is an RCM, was used for the downscaling simulations. The boundary conditions for SCALE-RM were constructed from the results of MRI-AGCM3.2S[19], which is an atmospheric general circulation model. See the Methods section for more details on the experimental setup. Three indices, indicated in Table 2, were used for the evaluation.

To simplify the subsequent discussion, we consider that the major factors affecting the precipitation change due to climatology and perturbation changes are roughly understood to be the thermodynamic and dynamic changes, respectively. This would be reasonable in at least the mid latitudes and high latitudes; for the climatology change, a previous study[9] already showed that the influence of thermodynamic change is larger than that of dynamic change, while for the perturbation change, the indirect thermodynamic influence due to slight changes in the disturbance track may be relatively low compared with the direct dynamic influence due to changes in the frequency, intensity and track of the disturbance in a limited area.

RAVE in the Present-DDS, Future-DDS and their difference are shown in Fig. 2a–c, respectively. RAVE in the future climate decreases from that in the present climate for most areas. Figure 2d, e show the changes in RAVE due to the changes in the climatology and perturbation, respectively. Changing the climatology slightly increases RAVE, especially for the northern area, while changing the perturbation accounts for a large part of the total decrease in RAVE. The areal mean of the RAVE change is summarised in Fig. 3a. The rate of increase in mean precipitation due to the changes in climatology is approximately 3% under the 3.6 °C warming of the surface temperature from the present climate; this value is consistent with those in previous studies[20,21]. On the other hand, the rate of decrease due to the changes in perturbation is ~14%; the reason for the large decrease is explained later. The change in RAVE for the expected climate change (Fig. 2f) has a similar distribution to that of the total mean precipitation change (Fig. 2c) in most areas. This means that the mean precipitation change can largely be explained by summing the influences of the climatology and perturbation changes. Figure 2g shows the nonlinear effect $\Delta cp$ between the changes in the two components. Although the magnitude of the nonlinear effect is small over most areas, it is comparable to the influence of the changes in the climatology. The relationship between the expected climate

change and $\Delta cp$ varies with the location. In terms of the areal mean, the nonlinear effect tends to enhance the expected climate change for RAVE because $(\Delta C_0 + \Delta P_0) \cdot \Delta cp > 0$, as shown in Fig. 3a.

To investigate whether the results of the mean precipitation would hold true for extreme precipitation, other precipitation indices, R1D and CDD, were also evaluated. Although R1D shows little change between the present and future climates (Fig. 3b), this result does not mean that the changes in each large-scale atmospheric component have no effect. The decrease in R1D due to the perturbation change is comparable to the increase due to the climatology change; the small change in R1D between the present and future climates is simply a consequence of cancelling the contributions from the two component changes. The nonlinear effect suppresses the expected climate change for R1D, because the expected climate change slightly increases and $(\Delta C_0 + \Delta P_0) \cdot \Delta cp < 0$. The total change in CDD increases as a result of the changes in both components (Fig. 3c). Because $(\Delta C_0 + \Delta P_0) \cdot \Delta cp > 0$, the nonlinear effect enhances the expected climate change in CDD. Thus, the magnitude and direction of the contributions of the climatology and perturbation depend on the target index. The role of the nonlinear effect also depends on the index. Its magnitude is relatively small but not negligible for the total change.

The changes in climatology increased both the mean and extreme precipitations. The increasing rate of the extreme precipitation (R1D) was larger than that of the mean precipitation (RAVE); this result is consistent with those of previous studies[9,21]. Regardless of the mean and extreme precipitation, increases due to the climatology change can be explained by the thermodynamic changes in the large-scale atmosphere, i.e. the precipitation increases due to the enriched water vapour in the warm atmosphere related to the Clausius–Clapeyron relationship.

On the other hand, the mean and extreme precipitations were decreased by the changes in perturbation. This decrease exceeded or was comparable to the increase due to the changes in climatology, as shown in Fig. 3a, b. The perturbation change is particularly characterised by the changes in the frequency and intensity of the disturbances causing precipitation. According to a cyclone analysis based on the detection method of Adachi and Kimura[22], the decrease in precipitation associated with typhoons and extratropical cyclones accounted for more than 95% of the precipitation decrease due to the perturbation change between Present-DDS and Pseudo-Perturb-DS, whereas changes in the precipitation due to weather events other than cyclones were quite small. Further analysis showed that the numbers of typhoons and extratropical cyclones affecting the studied region were projected to decrease by 40 and 15%, respectively, whereas the precipitation brought on by a single event, i.e. one typhoon or one extratropical cyclone, to the region remained almost unchanged between the two experiments. Thus, the precipitation decrease due to perturbation changes can be explained by the decreases in the numbers of typhoon and extratropical cyclone events in this case.

When an adaptation strategy is considered, evaluating not only the changes in the cumulative precipitation amount but also those in the frequency and intensity of precipitation is important. The former information would be useful for securing water resources, while the latter information can help reduce the risk of disaster.

From the viewpoint of reducing disaster risk, Fig. 4a shows the relationship between the precipitation intensity and changes in wet days. The total change in wet days between the present and future climates is small across all categories. However, the small total change is a result of the counteracting contributions of the three studied factors. The changes in the perturbation decrease the wet days for all categories, while the changes in the

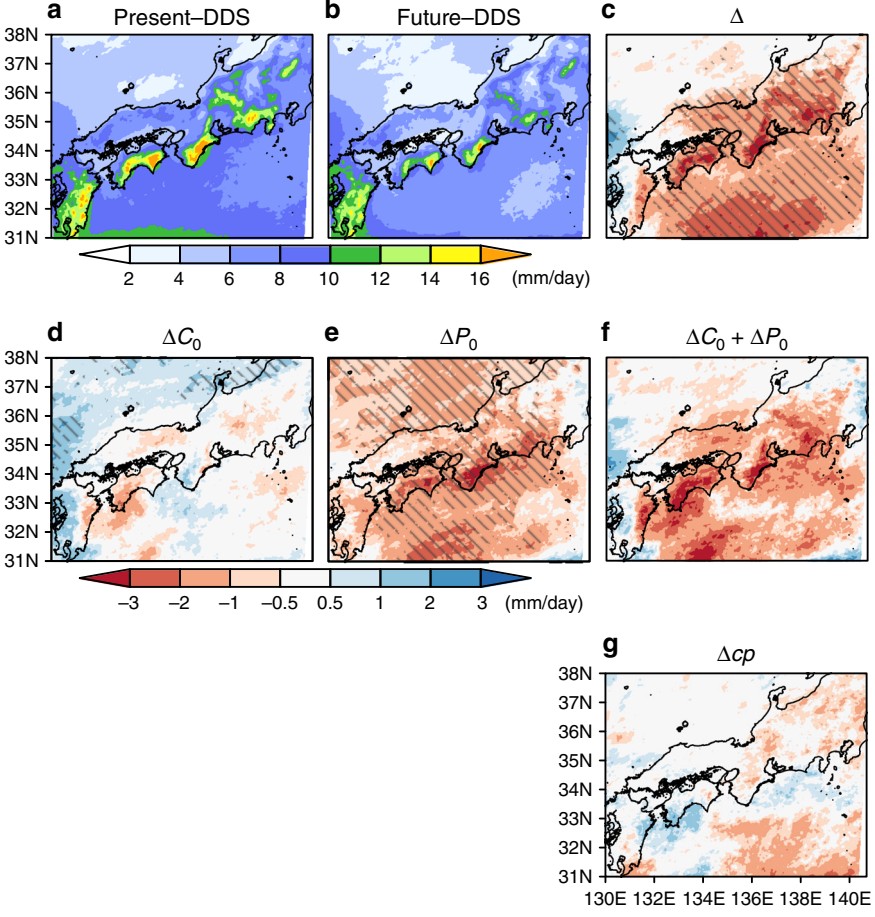

**Fig. 2** Spatial distribution of precipitation changes. Twenty-five-year averages of the daily precipitation in **a** Present-DDS, **b** Future-DDS and **c** the difference between Present-DDS and Future-DDS. The contributions of the climatology, perturbation and nonlinear effect are shown in **d**, **e**, **g**, respectively. The daily precipitation change estimated by the expected climate change is shown in **f**. The hatched regions in **c**–**e** represent the area in which the difference from Present-DDS is statistically significant with a 95% confidence level

climatology drastically increase the wet days in heavier precipitation. On the other hand, the nonlinear effect drastically decreases the wet days for extreme precipitation of more than 300 mm per day. That is, the nonlinear effect suppresses the precipitation increase due to the climatology change, especially for extreme precipitation. The increase in the frequency of extreme precipitation results from an increase due to the changes in the climatology outweighing the sum of the decrease due to the perturbation changes and counteraction from the nonlinear effect.

From the viewpoint of securing water resources, Fig. 4b shows the amount of cumulative precipitation against the precipitation intensity. Most of the precipitation amount results from weak precipitation of less than 100 mm per day. The cumulative precipitations in Future-DDS and Pseudo-Perturb-DS are only ~90% of that in Present-DDS because the frequency of weak precipitation of less than 100 mm per day decreases in Future-DDS and Pseudo-Perturb-DS compared to in Present-DDS, as shown in Fig. 4a. This means that the changes in the perturbation affect the frequency of weak precipitation, which changes the amount of precipitation.

We had presumed that the precipitation amount would be affected by climatology changes, whereas the intensity might rather be influenced by perturbation changes. However, the demonstrated results were different from our expectation. The cumulative precipitation was largely influenced by the changes in the perturbation, whereas the precipitation intensity was mainly affected by the changes in the climatology, at least in the demonstrated case.

## Discussion

Most of changes in the precipitation due to climatology changes can be explained by the thermodynamic change in the large-scale atmosphere associated with the Clausius–Clapeyron effect. On the other hand, the changes in climatology also include changes in the dynamic structure of the atmosphere, such as the atmospheric stability, as described above. Because the temperature increase due to global warming is greater in the upper troposphere, the vertical stability increases[23]. The climatology projected in MRI-AGCM3.2S also indicates such stabilisation, and the downscaled results in Pseudo-Clim-DS and Future-DDS reflect this tendency. Regarding the influence of changes in climatology on the precipitation intensity, the results from Pseudo-Clim-DS indicate that heavy precipitation was enhanced and weak precipitation was slightly suppressed (Fig. 4). The reduction of weak precipitation seems to be attributable to the stabilisation of the atmosphere in the future climate. To more quantitatively evaluate these two effects, further experiments using appropriate boundary conditions are required. As an example, Kröner et al.[24] proposed a downscaling approach to evaluate the separate contributions of vertically homogeneous warming and a vertical structure change in the temperature associated with changes in climatology to the regional climate change.

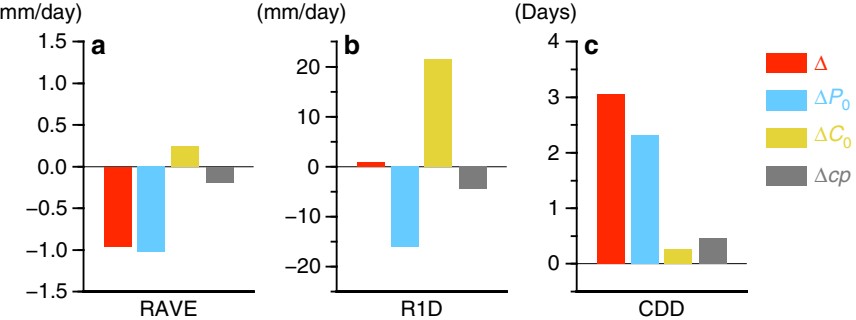

**Fig. 3** Changes in the precipitation indices for evaluation. Differences from the present climate in domain-averaged 25-year means of the **a** daily mean precipitation (RAVE), **b** maximum one-day precipitation (R1D) and **c** maximum number of consecutive dry days (CDD), respectively. $\Delta$ is the total change estimated by the DDSs (Future-DDS − Present-DDS). $\Delta P_0$ and $\Delta C_0$ indicate the contributions of the perturbation and climatology, respectively. $\Delta cp$ is the contribution of the nonlinear effect

If the changes in perturbation are considered along with those in climatology, the regional precipitation change can be determined by the balance of the thermodynamic effect in the mean state, the changes in the mean atmospheric structure, and changes in perturbation. Among these, the influence of the perturbation changes is strongly dependent on the location and size of an evaluated area; the smaller the size of the evaluated area is, the larger the influence due to the perturbation change becomes. The changes in perturbation are potentially crucial for the future regional climate.

The demonstrated results show a view regarding a debatable point whether the use of the pseudo-warming method is valid for projecting the future regional climate. The pseudo-warming method has the advantage of reducing the model bias in a GCM projection by using the reanalysis data for the present climate. However, the future downscaling projection is performed using the present perturbation[14,25,26]. There have been concerns that the pseudo-warming method may underestimate or overestimate extremes for the projected climate, because the activities of atmospheric disturbances, such as tropical and extratropical cyclones, are projected to change due to global warming in some models[27,28]. In our demonstration of the proposed procedure, perturbation changes have a large impact on the precipitation change. This means that the effect of the changes in perturbation cannot be negligible for the projection in areas where the frequency and intensity of large-scale disturbances are projected to change in the future climate. The demonstration further indicates that the nonlinear effect inhibits the intensification of precipitation due to climatology changes. This result implies the possibility that the pseudo-warming method overestimates the extreme precipitation events, at least in the area and season evaluated. Further evaluation of the physical meaning of the nonlinear effect is necessary for a greater understanding of the essence of the pseudo-warming method.

Although many studies have concentrated on projecting the future climate and evaluating the effect of climatology changes so far, the results of our procedure emphasise the importance of the symmetric treatment of changes in climatology and perturbation along with the nonlinear effect to precisely understand the regional climate projection. By considering the influence of each change and their nonlinear effects, the procedure could provide clues to the mechanisms that cause future regional climate change. To understand the mechanisms in more detail, it would be helpful to separate the factors affecting the regional climate and evaluate their contributions quantitatively; for example, thermodynamic and dynamic changes in climatology, those in perturbation and ground surface conditions such as land use.

In this study, only a pair of present and future climates estimated by a single GCM was used. However, there is uncertainty in future climate projections with GCMs because of imperfections regarding the emission scenario and model, amongst others. The uncertainties would be involved in both the climatology and perturbation of the future projections. By applying our procedure to multi-regional climate projections using multi-GCM projections, the procedure may also provide information on the characteristics of uncertainty in regional climate projections, such as which atmospheric component causes a spread in regional climate projections.

## Methods

**Regional climate model used in this study.** The regional model used for the downscale experiments is based on Scalable Computing for Advanced Library and Environment-Regional Model (SCALE-RM)[16–18] version 4.2.5. The calculation domains cover an area of $2520 \times 2520 \ km^2$ with a grid spacing of 7.5 km for the outer domain and an area of $1080 \times 960 \ km^2$ with a grid spacing of 2.5 km for the inner domain. In the vertical direction, the outer and inner domains have 36 and 60 layers, respectively. The calculation period for the downscaling experiments was from 31 May to 30 September during 1979–2003 for the present climate experiment and during 2075–2099 for the future climate experiment. This 4-month calculation in each year was divided into 31 runs, and each run was conducted every four days. The integration period of one run was five days, which consisted of one day for model spin-up and four days for analysis. The outputs from the last 4 days were connected sequentially and used for the analysis. Thus, the period from 1 June to 30 September of each year was analysed.

The large-scale climate data used for the initial and boundary conditions of the RCM were provided from 6-hourly data in the present and future climates estimated by MRI-AGCM3.2S[19], as explained in the next subsection. The following variables are used for the initial and boundary conditions: atmospheric temperature, geopotential height, pressure, specific humidity, zonal and meridional winds, surface pressure, skin temperature, sea surface temperature (SST), soil temperature and soil moisture. Note that the SST was given as a temporally averaged value in each integration period and then kept constant throughout a five-day simulation. The initial and boundary conditions for the Present-DDS and Future-DDS experiments are directly given from GCM outputs, whereas these conditions for the other two experiments are prepared according to the following procedure using GCM outputs. First, the climatology and perturbation were calculated for the present and future climate. The climatology of the large-scale atmospheric state was determined via two steps: (1) monthly means averaged over 25 years were prepared from May to October; (2) regarding these as the 15th day of each month, the climatology on each day was determined by their linear interpolation. Thus, the climatology varied from day to day. The perturbation was calculated by subtracting the obtained daily climatology from the 6-hourly data. Then, the initial and boundary conditions of the RCM were prepared by combining these climatology and perturbation according to the experimental design listed in Table 1, except for the specific humidity. The specific humidity is calculated from the temperature constructed by the procedure described above and the relative humidity. The relative humidity used for Pseudo-Clim-DS and Pseudo-Perturb-DS is assumed to depend on the climate of perturbation, namely, the relative humidity for these experiments were obtained from the present and future GCM simulations, respectively. The assumption that the relative humidity remains unchanged between present and future climates is common in the pseudo-warming method[24,25]. This assumption is based on reports that the change in relative humidity is projected to be quite small in future warmer climates[6,29]. The

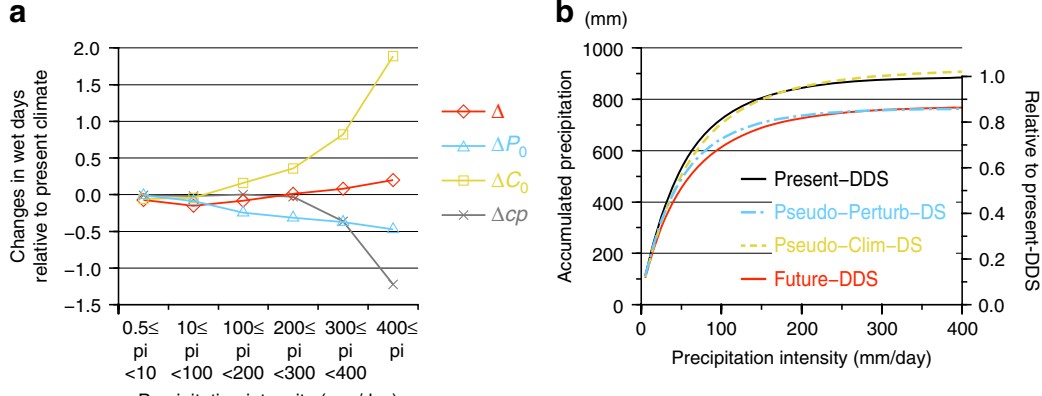

**Fig. 4** Changes in precipitation characteristics for different precipitation intensities. **a** Difference in the number of wet days from the present climate for each category of precipitation intensity and **b** the cumulative precipitation amount. The value in **a** is normalised by the number of wet days in Present-DDS for every intensity category. $\Delta$ is the total change estimated by the DDSs (Future-DDS − Present-DDS). $\Delta P_0$ and $\Delta C_0$ indicate the contributions of the perturbation and climatology, respectively. $\Delta cp$ is the contribution of the nonlinear effect. The right scale in **b** indicates the ratio to the accumulated precipitation of the present climate (Present-DDS)

advantage of using this assumption is to be able to avoid super-saturated and super-dried conditions.

The cloud microphysics and turbulence processes were calculated according to the six-class single-moment bulk scheme[30] and Mellor–Yamada Nakanishi–Niino level 2.5 TKE scheme[31], respectively. The radiation process was calculated with the MSTRN-X radiative scheme[32]. The concentrations of $CO_2$ and other greenhouse gases in the atmosphere used in the radiative scheme are taken from those for the scenario of representative concentration pathway (RCP) 8.5, except for Present-DDS. The land variables were solved with the five-layer heat diffusion and bucket model using surface flux parameterisation[33]. The fluxes from urban areas were calculated with the single-layer urban canopy model[34]. No nudging techniques were used in this simulation, because the integration period for each run was only 5 days.

**Present and future climate data with MRI-AGCM3.2S**. The climate data for the boundary conditions of SCALE-RM were calculated by MRI-AGCM3.2S[19]. MRI-AGCM3.2S is a high-resolution hydrostatic atmospheric GCM developed at the Meteorological Research Institute (MRI) for climate simulations. The SST forcing data for MRI-AGCM3.2S were taken from the Hadley Centre Sea Ice and SST data set version 1 (HadISST1)[35] for the present climate. The future climate was projected using the multi-model mean of 28 SST data projected by the Coupled Model Intercomparison Project Phase 5 (CMIP5) atmosphere-ocean coupled GCMs (CGCMs) under the RCP8.5 scenario[36].

**Code availability**. The regional model for the downscaling calculation in this study was constructed by using SCALE, which is freely available at https://scale.aics.riken.jp/download/index.html under the 2-Clause BSD license. The model code used is based on SCALE-RM version 4.2.5 including some improvements such as the usability of the microphysics scheme. The model code and the set of configuration parameters are available from the corresponding author upon reasonable request. The pre-process code that generates the boundary conditions for SCALE-RM from the MRI-AGCM3.2S data and the post analysis code used in this study are also available from the corresponding author.

**Data availability**. The downscaling data by SCALE-RM are deposited in local storage at RIKEN/AICS. It is available from the corresponding author upon reasonable request. The MRI-AGCM3.2S data were provided to the authors by MRI under a given condition that the authors can use the MRI-AGCM3.2S data only for the current and related studies. The data is not publicly available. However, upon reasonable request with regards to this study, the data are provided from MRI with the permission of MRI. In this case, contact person is the corresponding author.

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

## Acknowledgements

This study was supported by the Foundation for Computational Science (FOCUS) establishing the Supercomputing Center of Excellence and JST CREST (Grant Number JPMJCR1312), Japan. The results of this study were obtained using the computational resources of the K computer at RIKEN/AICS. We thank Drs I. Takayabu, R. Mizuta and O. Arakawa for providing the MRI-AGCM3.2S data. The climate simulations with MRI-AGCM3.2S were carried out by the MRI under the Program for Risk Information on Climate Change (SOUSEI) of the Ministry of Education, Culture, Sports, Science and Technology (MEXT) of Japan. We thank the two anonymous reviewers for their fruitful discussions on improving this paper.

## Author contributions

S.A.A., S.N., Y.K. and H.T. designed the research; S.A.A., S.N., T.Y., K.A. and H.Y. prepared the boundary data and performed the simulations; S.A.A., S.N., R.Y. and T.Y. analysed the data; and mainly S.A.A. and H.T. wrote the paper.

## Additional information

**Competing interests:** The authors declare no competing financial interests.

