## [Peer Review File · Nature Communications]

Reviewer #1 (Remarks to the Author):

This work proposes a new method of decomposing climate change simulations produced by limited area models for a better understanding of the roles of changes in mean climate, changes in variability and their interactions. Although the method is interesting and looks useful, I'm not yet fully convinced of the usefulness.

My major concern is the lack of physical interpretation of the decomposed fields in the demonstrated example. The decomposed fields of changes in mean precipitation (Fig. 2), for example, show interesting patterns. Although the characteristics of the patterns are clearly described and how the method works are interpreted, I think the authors need to describe physical interpretation as well, like why the decrease in precipitation due to changes in perturbation dominates, referring to atmospheric mechanisms that cause precipitation over the area and the season studied. Without such interpretation, readers can't judge whether this method is physically meaningful or is just a technical argument. The same is true for the Figs. 3 and 4. Describing those interpretations, however, may lengthen the manuscript. The authors could try to manage the text to fit in the length limit, but if they find it too difficult, that might mean this paper fits better to a full paper rather than to a letter-type publication.

My other comments follow:

1. I'm not comfortable with the use of expressions "quantitative" and "qualitative" changes in precipitation. Changes in perturbations can naturally be characterized by quantitative numbers like frequency of certain events or amplitude of variabilities, thus not really "qualitative".

2. The description of experimental design doesn't seem fully clear:

Line 232: "the monthly average for each day": Does it mean monthly running mean centered around each day (thus changes day by day) or monthly mean for a given month (temporally constant field throughout the month)?

Line 235-236: The frequency of time-slice experiments conducted is not clear. Are they conducted for every 4 days so that you can have a continuous analysis period, every 5 days and you have 1 day gap in every 5 days as the days used for analysis, even less frequent with more gaps, or much more frequent with overlapping days for analysis?

3. The result of this study seems to have an implication for the pseudo global warming method. Namely, the dominance of perturbation component in the simulated changes in precipitation demonstrated here seems to imply that the pseudo global warming method, in which perturbation is not altered from reanalysis data, can't capture the dominant component of precipitation change. If this is true, it would mean the pseudo global warming method produces unreliable results at least for the demonstrated area and season. I might misunderstand it, but I want to see some discussion on this point.

4. The notation "Rave" looks like a word. It would be better to show "ave" in suffix rather than just lower case.

End of review

Reviewer 2

The authors present results from regional climate model simulations. The simulations are designed to give insight into the reliability of so-called “pseudo global warming” simulations, in which the climatological mean global warming signal of a GCM simulation is simply added to reanalysis data to produce boundary conditions for a regional model simulation for future climate. The pseudo global warming method is popular because of its simplicity and because of the built-in GCM bias correction, but there is concern that the resulting answers are not entirely correct as potentially important changes in transient variability (e.g., frequency and intensity of storms) are neglected.

Although this study is somewhat technical in nature and somewhat limited in scope (one specific model, one specific region, one specific variable), I find its results new, interesting, and relevant to anybody who is concerned with regional climate modeling. I also do not find any major flaws, and therefore recommend publishing the manuscript with major revisions, according my comments below.

Main comments:

148: I was surprised when seeing Fig. 2. Changes in mean temperature are expected to lead to considerable changes in precipitation because of the non-linear relationship between temperature and saturation vapor pressure (Clausius-Clapeyron equation). I few comments why this is here apparently not the case are warranted. Maybe, also an analysis of how the transients (perturbations) are changing over the study region would be nice. In other words, how does the frequency and intensity of storms change over the study region in order to produce most of the precipitation changes? Similarly, how does the mean climate over the region change, and why does this not influence precipitation in major ways. Also, is the result in Fig. 2 dependent on the study region, or do you expect that changes in the mean are almost everywhere equally unimportant?

233-236: I got really confused from reading the methodology. “5-day time slice experiments”?. So, these experiments are not continuous simulations? Much more detail is needed to understand this. What is the initial condition for each day’s “time slice”? How long does it take the model to equilibrate to the new initial condition, much longer than just 5 days, I think. Explain in more detail!

Minor comments:

Title: I find the “climatology, perturbation, and their interaction” in the title (and also later in the text) a bit unprecise and perhaps even misleading. In my opinion, this is about “Contributions of changes in the mean, changes in transients, and resulting non-linearities to regional climate change”.

Abstract: It is a bit hard to understand, in particular line 8, where “three contributions to regional climate change” are mentioned. From the previous text it is not entirely clear which three contributions are meant.

30: I do not entirely agree. Dynamic changes may also project on changes in the mean. For example, if the frequency or intensity of storms changes this will also project on the mean.

40-45: See my previous comment. Also, the time scale discussion is a bit awkward. Thermodynamic changes, which I agree are slow, can impact the rainfall intensity of a storm, which is a fast process. You say that “dynamic change corresponds to a change at a relatively short time scale”. I think this all does not really make sense here. Rainfall events are always short in nature because they are produced by storms, hence dynamics. But this is all irrelevant. I recommend removing this entire discussion about fast and slow.

45: Climatology are perturbation are not very precise expression here. I would prefer “changes in the mean ($\langle p \rangle \rightarrow \langle f \rangle$) and changes in transients ($p' \rightarrow f'$)”.

52-69: I find this literature review much too long and detailed. Just a few facts, that should be enough.

72: Abbreviations. I think the paper use too many, too complicated, and unfamiliar abbreviations (DDS, PDDS, FDDS, PCCDS, PPCDS, etc.), which are difficult to remember. Please avoid.

116-137 : I think this discussion is unnecessarily complicated. Delta-cp is simply the residual of the total response that cannot be explained by the simple linear addition of the two individual effects (Delta-C, Delta-P). Usually this is considered to be due to non-linearities of the system, and I would state it like that. The word “interaction” is misleading.

Table 2 and text: The naming of Rave is awkward. Can you put it in all capital letters? Also, in meteorology, P is usually used as variable name for rain. I recommend: P, P1D, and CDD (but not sure what it stands for).

Figure 1: This figure is a nice idea, but it is too complex and hard to understand. Please stick to the most important aspects and remove extra information.

223: Method. You need to explain in more detail how the BCs for the individual simulations are generated. For example, how do you calculate $\langle p \rangle$, $\langle f \rangle$, p' , and f' , and which model quantities are included. Do you use spectral nudging in your model?

Reply to Reviewer #1's comments:

This work proposes a new method of decomposing climate change simulations produced by limited area models for a better understanding of the roles of changes in mean climate, changes in variability and their interactions. Although the method is interesting and looks useful, I'm not yet fully convinced of the usefulness.

[Reply]

We are grateful to you for careful reading and evaluation of our manuscript. We considered your concerns and replied to all of your comments as precisely as possible. As a major revision, we added physical interpretations of the demonstrated results, and we believe the usefulness of our method is highlighted in the revised manuscript. Before reading our reply to your individual comments, please note that the names of the experiments have been changed from the previous manuscript. This is a request from another reviewer to avoid confusion on the reader's part.

Previous manuscript	New manuscript	Description of experiment
PDDS	Present-DDS	Present climate direct downscaling
FDDS	Future-DDS	Future climate direct downscaling
PCCDS	Pseudo-Clim-DS	Pseudo climatology downscaling
PPCDS	Pseudo-Perturb-DS	Pseudo perturbation downscaling

Here, DDS is a general abbreviation for direct (dynamical) downscaling.

My major concern is the lack of physical interpretation of the decomposed fields in the demonstrated example. The decomposed fields of changes in mean precipitation (Fig. 2), for example, show interesting patterns. Although the characteristics of the patterns are clearly described and how the method works are interpreted, I think the authors need to describe physical interpretation as well, like why the decrease in precipitation due to changes in perturbation dominates, referring to atmospheric mechanisms that cause precipitation over the area and the season studied. Without such interpretation, readers can't judge whether this method is physically meaningful or is just a technical argument. The same is true for the Figs. 3 and 4. Describing those interpretations, however, may lengthen the manuscript. The authors could try to manage the text to fit in the length limit, but if they

find it too difficult, that might mean this paper fits better to a full paper rather than to a letter-type publication.

[Reply]

Thank you very much for this valuable comment. We understand your concern. The physical interpretation of demonstrated results would be helpful for readers to understand the advantage of this method. According to your comment, we added the physical interpretation of the demonstrated results and our view on following four points:

- 1) Why does the precipitation decrease due to the changes in perturbation in association with the atmospheric mechanism causing precipitation over the target region and season?
- 2) How do the changes in climatology affect the precipitation in the target region?
- 3) Why was the influence of perturbation change on the precipitation dominant over that of climatology change in this region?
- 4) Is our procedure just a technical argument, or can it provide physical meaning?

The summary of our reply to each point is described with **bold face and underlined** in the first paragraph under each question.

1. Why does the precipitation decrease due to the changes in perturbation? We conclude that the drastic precipitation decrease due to the changes in perturbation is attributable to the decreases in the number of typhoons and extratropical cyclones passing through the target area. A detailed analysis is as follows:

From the phenomenological view, the precipitation over the demonstrated area and season is mainly brought on by four types of weather events: typhoons, extratropical cyclones, seasonal stationary fronts (known as Baiu fronts), and local rainfall due to convective instability.

In order to investigate which event-type brought on the drastic precipitation decrease, we extracted the precipitation associated with typhoons and extratropical cyclones. We picked up tropical and extratropical cyclones by applying the cyclone detection method of Adachi and Kimura (2007) to Pseudo-Perturb-DS and Present-DDS; the cyclones whose genesis were south of 25N were regarded as tropical cyclones (typhoons), whereas the remaining cyclones were regarded as extratropical cyclones. The precipitation associated with these cyclones were defined as precipitation within 1000 km from its centre point. As a result, we found that the precipitation changes

associated with typhoons and extratropical cyclones account for 63% and 34% of the precipitation decrease due to perturbation change, ΔP_0 , respectively, while the changes in other events, including seasonal stationary front and local rainfall due to convective instability, hardly contributed to ΔP_0 .

To further investigate the main cause of the precipitation decrease associated with cyclones, we analysed changes in intensity and frequency for each typhoon and extratropical cyclone. In terms of frequency, the numbers of detected typhoons and extratropical cyclones affecting to the target region decreased by 40% and 15%, respectively. In terms of intensity, we investigated the precipitation amount per cyclone event to evaluate the influence of cyclones on local precipitation. Note that the obtained values do not exactly represent the intensities of cyclones themselves but rather their impacts on the target region for precipitation. The impact per event on the precipitation varied little between the two experiments for both typhoons and extratropical cyclones.

We revised the manuscript as follows:

p.7, l.137:

“The Pacific side of western Japan has high rainfall in the summer season associated with typhoons, extratropical cyclones, seasonal stationary fronts, and local rainfall due to convective instability.”

p.8, l.179:

“On the other hand, the mean and extreme precipitation were decreased by the changes in perturbation. This decrease exceeded or was comparable to the increase due to the change in climatology as shown in Figs. 3a and b. The perturbation change is particularly characterised by the changes in frequency and intensity of disturbances causing precipitation. According to cyclone analysis based on the detection method of Adachi and Kimura (2007), the decrease in precipitation associated with typhoons and extratropical cyclones accounted for more than 95% of the precipitation decrease due to the perturbation change between Present-DDS and Pseudo-Perturb-DS, whereas changes in precipitation due to weather events other than cyclones was little. Further analysis showed that the numbers of typhoons and extratropical cyclones affecting to the studied region were projected to decrease by 40% and 15%, respectively, whereas the precipitation brought on by a single event, i.e. one typhoon or one extratropical cyclone, to the region remained almost unchanged between the two experiments. Thus, the precipitation decrease due to perturbation changes can be explained by the decreases in the numbers of typhoon and extratropical cyclone events in this case.”

2. How do the changes in climatology affect the precipitation in the target region?

The changes in mean precipitation due to the climatology changes are determined by the balance between the thermodynamic effect associated with the Clausius-Clapeyron relationship and changes in vertical stabilization. If we focus on the precipitation change in different categories of precipitation intensity (Fig. 4a), heavier rainfall is enhanced by the dominance of Clausius-Clapeyron effect, whereas weaker rainfall is suppressed by the dominance of the stabilisation effect.

The mean temperature increase associated with the changes in climatology enriches the water vapour in the atmosphere via the Clausius-Clapeyron relationship; this is expected to increase precipitation. On the other hand, the vertical stability increases because of the larger warming in the upper troposphere (Bonny et al., 2006); this is expected to suppress precipitation. It is considered that the balance between them determines the mean and extreme precipitation change. We confirmed the above by using Pseudo-Clim-DS and Present-DDS in our demonstrated results.

We added an interpretation of the influence of changes in climatology in the manuscript as follows:

p.7, l.148:

“The increasing rate of mean precipitation due to the changes in climatology is about 3% under the 3.6°C warming of surface temperature from the present climate; this value is consistent with those in previous studies (Held and Soden, 2006; Pall et al., 2007).”

p.8, l.173:

“The changes in climatology increased both the mean and extreme precipitation. The increasing rate in extreme precipitation (R1D) was larger than that in mean precipitation (RAVE); this result was consistent with those of previous studies (Emori and Brown, 2005; Pall et al., 2007). Regardless of mean and extreme precipitation, changes due to the climatology change can be explained by the thermodynamic change in the large-scale atmosphere, i.e. the precipitation increases due to the enriched water vapour in the warm atmosphere related to the Clausius-Clapeyron relationship.”

p.10, l.221:

“Most changes in the precipitation due to climatology changes can be explained by the thermodynamic change in the large-scale atmosphere associated with the Clausius-Clapeyron effect. On the other hand, the changes in climatology also include

changes in the dynamic structure of the atmosphere, such as atmospheric stability. Because the temperature increase due to global warming is greater in the upper troposphere, the vertical stability increases (Bonny et al. 2006). The climatology projected in MRI-AGCM3.2 also indicates such stabilization, and the downscaled results from Pseudo-Clim-DS and Future-DDS inherited this aspect. Regarding the influence of changes in climatology on precipitation intensity, the results from the Pseudo-Clim-DS indicate that heavy precipitation was enhanced and weak precipitation was slightly suppressed. The reduction of weak precipitation seems to be attributable to the stabilization of the atmosphere in the future climate.”

3. Why was the influence of perturbation change on the precipitation dominant over that of climatology change in this region?

In the case of our demonstration, the decreases in mean precipitation associated with the decrease in the number of cyclones surpassed the precipitation increases resulting from the balance between the Clausius-Clapeyron effect and the stabilization in warming climate. Whether climatology or perturbation changes is dominant depends on location and season as well as the size of an evaluated region.

In general, the regional climate is strongly affected by (i) how the synoptic disturbances pass through the target region and (ii) how large the evaluated region is. In the region where synoptic disturbances bring on a substantial amount of precipitation, the impact of perturbation change on the regional precipitation strongly depends on the abovementioned two conditions. That is, the degree of influence of changes in perturbation largely varies with location and size of the target area. For example, there is a possibility that disturbances, causing most of the precipitation in the present climate in an area, hardly pass through the area in the future climate.

We added the sentences below to the revised manuscript:

p.10, l.232:

“If the changes in perturbation are considered along with those in climatology, the regional precipitation change can be determined by the balance of the thermodynamic effect, the changes in mean atmospheric structure, and changes in perturbation characteristics. Among these, the influence of perturbation changes is strongly dependent on the location and size of an evaluated area; the smaller the size of the evaluated area is, the larger the influence due to perturbation change becomes. The changes in perturbation are potentially crucial for future regional climate.”

p.10, l.245:

“In our demonstration of the proposed procedure, perturbation changes have a large impact on the precipitation change. This means that the effect of the changes in perturbation cannot be negligible for the projection in an area where the frequency and intensity of large-scale disturbances are projected to change in the future climate.”

Please note that the results in the demonstration strongly depend on the used GCM, because the projection would include uncertainty due to model bias as well as emission scenario. We consequently added the below sentences in the manuscript:

p.11, l.260:

“In this study, only a pair of present and future climates estimated by a single GCM was used. However, there is uncertainty in future climate projections with GCMs because of imperfections regarding the emission scenario and model, amongst others. The uncertainties would be involved in both the climatology and perturbation components of future projections. By applying the procedure to multi-regional climate projections using multi-GCM projections, the procedure may also provide information on the characteristics of uncertainty in regional climate projections, such as which atmospheric component causes a spread in regional climate projections.”

4. Is our procedure just a technical argument, or can it provide physical meaning?

Our method emphasises the importance of the symmetric treatment of changes in climatology and perturbation. By considering each influence and their nonlinear effect, we advance understanding of the mechanism of regional climate change. Thus, we believe that our proposed procedure is not just a technical argument but also leads to giant step forward to a better understanding of regional climate change.

The novelty of our proposed method is to enable the quantitative evaluation of each effect – changes in climatology and perturbation as well as the nonlinear effect between them. The mathematical meaning is clear and is described under the heading “Principal Concept and Procedure” in the manuscript. We show that the method aids the physical interpretations under “Demonstration of the Proposed Procedure” and “Discussion” in the revised manuscript, for example, the impacts by different mechanisms of climatology and perturbation on regional precipitation change, and the effect of

nonlinearity between the two changes in comparison with the expected climate change (linear combination of two changes).

Many studies have concentrated on projecting the future climate and evaluating the effect of changes in climatology. However, to fully understand the mechanism causing regional climate change, it is important to consider not only one side, that is, climatology change, but also perturbation change. Our thoughts described above are expressed as follows in the manuscript:

p.11, l.254:

“Although many studies have concentrated on projecting the future climate and evaluating the effect of climatology changes so far, the results of our procedure emphasise the importance of the symmetric treatment of changes in climatology and perturbation along with the nonlinear effect to precisely understand the regional climate projection. By considering the influence of each change and their nonlinear effects, the procedure could provide clues to mechanisms that cause future regional climate change.”

A further detailed analysis is not within the scope of this paper, because its main purpose is the demonstration of the proposed procedure. We consider further evaluation of the physical meaning of the results from this procedure to be necessary in future studies by applying it to any other regions and seasons.

References:

- Adachi, S. and Kimura F. A 36-year Climatology of surface cyclogenesis in East Asia using high-resolution reanalysis data. *Sci. Online Lett. Atmos. (SOLA)* **3**, 113–116 (2007). doi:10.2151/sola.2007-029.
- Bony S. *et al.* How well do we understand and evaluate climate change feedback processes? *J. Climate* **19**, 3445–3482 (2006).
- Held, I.M. and B.J. Soden. Robust responses of the hydrological cycle to global warming. *J. Climate* **19**, 5686–5699 (2006). doi:10.1175/JCLI3990.1.
- Pall, P., Allen, M.R. and Stone, D.A. Testing the Clausius–Clapeyron constraint on changes in extreme precipitation under CO₂ warming. *Clim. Dyn.*, **28**, 351 (2007). doi:10.1007/s00382-006-0180-2.

My other comments follow:

1. I'm not comfortable with the use of expressions “quantitative” and “qualitative” changes in

precipitation. Changes in perturbations can naturally be characterized by quantitative numbers like frequency of certain events or amplitude of variabilities, thus not really “qualitative”.

[Reply]

We agree with this point. We revised the manuscript, without using the words “quantitative” and “qualitative”.

In the Introduction, we revised the manuscript as follows: (Note that the sentences have been modified from the previous manuscript in response to a comment from another reviewer.)

p.3, l.29:

“We presume that the precipitation amount is mainly affected by thermodynamic changes, whereas the intensity and frequency of precipitation might rather be influenced by dynamic changes.”

In the “Demonstration of the Proposed Procedure” section, we revised the manuscript as follows:

p.9, l.193:

“When an adaptation strategy is considered, evaluating not only the changes in cumulative precipitation amount but also those in frequency and intensity of precipitation is important. The former information would be useful for securing water resources, while the latter information can help to reduce the risk of disaster.”

p.9, l.197:

“From the viewpoint of reducing disaster risk, Fig. 4a shows the relationship between the changes in wet days and precipitation intensity.”

p.9, l.208:

“From the viewpoint of securing water resources, Fig. 4b shows the amount of cumulative precipitation against the precipitation intensity.”

p.9, l.215:

“We had presumed that the precipitation amount would be affected by climatology changes, whereas the intensity might rather be influenced by perturbation changes. However, the demonstrated results were different from our expectation. The cumulative precipitation was largely influenced by the changes in the perturbation, whereas the precipitation intensity was mainly affected by the changes in the climatology, at least in the demonstrated case.”

2. The description of experimental design doesn't seem fully clear:

Line 232: "the monthly average for each day": Does it mean monthly running mean centered around each day (thus changes day by day) or monthly mean for a given month (temporally constant field throughout the month)?

[Reply]

Thank you for pointing this out. The detailed procedure of making the climatology was added in the revised manuscript:

p.12, l.282:

"The climatology of the large-scale atmospheric state was determined via two steps. First, monthly means averaged over 25 years were prepared from May to October. Second, regarding these as the 15th day of each month, the climatology on each day was determined by their linear interpolation. Thus, the climatology varied from day to day."

Line 235-236: The frequency of time-slice experiments conducted is not clear. Are they conducted for every 4 days so that you can have a continuous analysis period, every 5 days and you have 1 day gap in every 5 days as the days used for analysis, even less frequent with more gaps, or much more frequent with overlapping days for analysis?

[Reply]

Thank you for pointing out the necessity of a detailed explanation of the time-slice experiment. Figure R1_1 shows the schematic diagram of this experiment. Each run had a 5-day integration period, which consisted of the first day for model spin-up and the last four days for analysis. For the calculation of the 4-month climate, 31 runs were conducted. We used a continuous analysis data from 1 June to 30 September by sequentially connecting all data. We have revised the text to clarify the time-slice experiment as follows:

p.12, l.275:

"The calculation was conducted as a time-slice experiment. The calculation for four months in each year was divided into 31 runs, and each run was conducted every four days. The integration period of one run was five days, which consisted of one day for model spin-up and four days for analysis. The outputs in last four days were connected sequentially and used for analysis."

Figure R1_1:
Schematic diagram of time-slice experiment, which explains the relationship between integration period and analysis data.

You may be concerned about whether the spin-up time was enough. Figure R1_2 indicates the composite of precipitation output during the 5-day integration for each experiment. The data were averaged in the analysis domain and all time-slice runs (31 runs x 25 years = 775 runs per experiment). There are no apparent biases in precipitation for the integration period of more than 24 h. According to this evaluation, we confirmed that the spin-up period of 1 day is enough, at least for this study.

Figure R1_2: Relationship between precipitation output and integration time. The data were averaged in the analysis domain and all time-slice runs (31 runs x 25 years = 775 runs per experiment).

3. The result of this study seems to have an implication for the pseudo global warming method. Namely, the dominance of perturbation component in the simulated changes in precipitation demonstrated here seems to imply that the pseudo global warming method, in which perturbation is not altered from reanalysis data, can't capture the dominant component of precipitation change. If this is true, it would mean the pseudo global warming method produces unreliable results at least for the demonstrated area and season. I might misunderstand it, but I want to see some discussion on this point.

[Reply]

In our view, the pseudo-warming method is not sufficient for the purpose of projecting the future regional climate in a region where the frequency and intensity of large-scale perturbations, such as cyclones, will change in the future climate. However, as we showed in the demonstration of the Pseudo-Clim-DS experiment, the pseudo-warming method is meaningful as a sensitivity study for extracting a response of a certain perturbation condition to the different mean state. It should be remembered that extreme events are strongly affected by changes in perturbation and climatology as well as their nonlinear effect. To enhance understanding of the results from the pseudo-warming method, the meaning of the nonlinear effect should be considered in future studies.

In our demonstration, the changes in perturbation have a large impact on future precipitation in the demonstrated area and season. However, there is much uncertainty in the projections of perturbation by GCMs as described above. It is also true that the climatology projected by GCMs includes uncertainty. We can evaluate the sources of uncertainty in the projections by applying the proposed method to multi-GCM projections.

Based on the above scope, we modified the last two paragraphs in the "Discussion" section as follows:

p.10, l.238:

"The demonstrated results show a view regarding a debatable point whether the use of the pseudo-warming method is valid for projecting future regional climate. The pseudo-warming method has the advantage of reducing the model bias in a GCM projection by using the reanalysis data for the present climate. However, the future downscaling projection is performed using the present perturbation (Sato et al. 2007; Kawase et al. 2008; Adachi et al. 2012). There have been concerns that the pseudo-warming method may under- or overestimate extremes for the projected

climate, because the activities of atmospheric disturbances, such as tropical and extratropical cyclones, are projected to change due to global warming in some models (Meehl and Tebaldi, 2004; Yamada et al. 2010). In our demonstration of the proposed procedure, perturbation changes have a large impact on the precipitation change. This means that the effect of the changes in perturbation cannot be negligible for the projection in an area where the frequency and intensity of large-scale disturbances are projected to change in the future climate. The demonstration further indicates that the nonlinear effect inhibits the intensification of precipitation due to climatology changes. This result implies the possibility that the pseudo-warming method overestimates the extreme precipitation events, at least in the area and season evaluated. Further evaluation on the physical meaning of the nonlinear effect is necessary for greater understanding of the essence of the pseudo-warming method in future.

Although many studies have concentrated on projecting the future climate and evaluating the effect of climatology changes so far, the results of our procedure emphasise the importance of the symmetric treatment of changes in climatology and perturbation along with the nonlinear effect to precisely understand the regional climate projection. By considering the influence of each change and their nonlinear effects, the procedure could provide clues to mechanisms that cause future regional climate change. ”

p.11, l.263:

“By applying the procedure to multi-regional climate projections using multi-GCM projections, the procedure may also provide information on the characteristics of uncertainty in regional climate projections, such as which atmospheric component causes a spread in regional climate projections.”

4. The notation “Rave” looks like a word. It would be better to show “ave” in suffix rather than just lower case.

[Reply]

Thank you for your suggestion. This was also pointed out by another reviewer, but he/she suggested a different idea. We decided to show the word in capitals, that is, RAVE, for readability.

Reply to Reviewer #2's comments

The authors present results from regional climate model simulations. The simulations are designed to give insight into the reliability of so-called “pseudo global warming” simulations, in which the climatological mean global warming signal of a GCM simulation is simply added to reanalysis data to produce boundary conditions for a regional model simulation for future climate. The pseudo global warming method is popular because of its simplicity and because of the built-in GCM bias correction, but there is concern that the resulting answers are not entirely correct as potentially important changes in transient variability (e.g., frequency and intensity of storms) are neglected.

Although this study is somewhat technical in nature and somewhat limited in scope (one specific model, one specific region, one specific variable), I find its results new, interesting, and relevant to anybody who is concerned with regional climate modeling. I also do not find any major flaws, and therefore recommend publishing the manuscript with major revisions, according my comments below.

[Reply]

We are grateful to you for carefully reading and evaluating our manuscript. We considered your concerns and replied to all comments as precisely as possible. We believe that the new manuscript is improved by our carefully considering your valuable comments.

Before reading our replies to your individual comments, please note that the names of the experiments have been changed in the revised manuscript and this reply, following your suggestions as below.

Previous manuscript	New manuscript	Description of experiment
PDDS	Present-DDS	Present climate direct downscaling
FDDS	Future-DDS	Future climate direct downscaling
PCCDS	Pseudo-Clim-DS	Pseudo climatology downscaling
PPCDS	Pseudo-Perturb-DS	Pseudo perturbation downscaling

Here, DDS is a general abbreviation for direct (dynamical) downscaling.

Main comments:

148: I was surprised when seeing Fig. 2. Changes in mean temperature are expected to lead to considerable changes in precipitation because of the non-linear relationship between temperature and saturation vapor pressure (Clausius-Clapeyron equation). I few comments why this is here apparently not the case are warranted. Maybe, also an analysis of how the transients (perturbations) are changing over the study region would be nice. In other words, how does the frequency and intensity of storms change over the study region in order to produce most of the precipitation changes? Similarly, how does the mean climate over the region change, and why does this not influence precipitation in major ways. Also, is the result in Fig. 2 dependent on the study region, or do you expect that changes in the mean are almost everywhere equally unimportant?

[Reply]

We appreciate your comment. We think that all your questions are valid. According to your specific questions, we would like to add the physical interpretation of our demonstrated results, focusing on the following four points:

- 1) How do the changes in frequency and intensity of storms contribute to the precipitation change over the target region?
- 2) How do the changes in climatology affect the precipitation change referring to the Clausius-Clapeyron relationship?
- 3) Why do the changes in climatology not dominantly influence the mean precipitation?
- 4) Is it possible to apply the consequences of the demonstrated results everywhere?

A summary of our reply to each point is described in **bold face and underlined** in the first paragraph under each question.

1. How do the changes in frequency and intensity of storms contribute to the precipitation change over the target region?

We conclude that the drastic precipitation decrease due to the changes in perturbation is attributed to decreases in the number of typhoons and extratropical cyclones passing through the target area. A detailed analysis is as follows:

From the phenomenological view, the precipitation over the demonstrated area and season is mainly brought on from four types of events: typhoons, extratropical cyclones, seasonal stationary fronts (known as Baiu fronts), and local rainfall due to convective instability.

In order to investigate which event-type brought on the drastic precipitation decrease, we extracted the precipitation associated with typhoons and extratropical cyclones. We picked up tropical and extratropical cyclones by applying the cyclone-detection method of Adachi and Kimura (2007) to Pseudo-Perturb-DS and Present-DDS; the cyclones whose genesis were south of 25N were regarded as tropical cyclones (typhoons), whereas the remaining cyclones were regarded as extratropical cyclones. The precipitation associated with these cyclones was defined as precipitation within 1000 km from their centre point. As a result, we found that the precipitation changes associated with typhoons and extratropical cyclones account for 63% and 34% of the precipitation decrease due to perturbation change, ΔP_0 , respectively, while the changes in other events, including seasonal stationary front and local rainfall due to convective instability, hardly contributed to ΔP_0 .

To further investigate the main cause of the precipitation decrease associated with cyclones, we analysed changes in intensity and frequency for each typhoon and extratropical cyclone. In terms of frequency, the numbers of detected typhoons and extratropical cyclones affecting the target region decreased by 40% and 15%, respectively. In terms of intensity, we investigated the precipitation amount per cyclone event to evaluate the influence of cyclones on local precipitation. Note that the obtained values do not exactly represent the intensities of the cyclones themselves but rather their impacts on the target region for precipitation. The impact per event on the precipitation did not vary much between two experiments for both typhoon and extratropical cyclone.

We added a summary of above analysis to the manuscript as follows:

p.7, l.137:

“The Pacific side of western Japan has high rainfall in the summer season associated with typhoons, extratropical cyclones, seasonal stationary fronts, and local rainfall due to convective instability.”

p.8, l.179:

“On the other hand, the mean and extreme precipitation were decreased by the changes in perturbation. This decrease exceeded or was comparable to the increase due to the change in climatology as shown in Figs. 3a and b. The perturbation change is particularly characterised by the changes in frequency and intensity of disturbances causing precipitation. According to cyclone analysis based on the detection method of Adachi and Kimura (2007), the decrease in precipitation associated with typhoons and extratropical cyclones accounted for more than 95% of the precipitation decrease due to the perturbation change between Present-DDS and

Pseudo-Perturb-DS, whereas changes in precipitation due to weather events other than cyclones was little. Further analysis showed that the numbers of typhoons and extratropical cyclones affecting to the studied region were projected to decrease by 40% and 15%, respectively, whereas the precipitation brought on by a single event, i.e. one typhoon or one extratropical cyclone, to the region remained almost unchanged between the two experiments. Thus, the precipitation decrease due to perturbation changes can be explained by the decreases in the numbers of typhoon and extratropical cyclone events in this case.”

2. How do the changes in climatology affect the precipitation change in the target region?

The changes in mean precipitation due to the climatology changes are determined by the balance between the thermodynamic effect associated with the Clausius-Clapeyron relationship and changes in vertical stabilization. If we focus on the precipitation change in different categories of precipitation intensity (Fig. 4a), heavier rainfall is enhanced by the dominance of the Clausius-Clapeyron effect, while weaker rainfall is suppressed by the dominance of the stabilization effect.

The mean temperature increase associated with the changes in climatology enriches the water vapour in the atmosphere via the Clausius-Clapeyron relationship; this is expected to increase precipitation. On the other hand, the vertical stability increases in the future climate because of the larger warming in the upper troposphere (Bonny et al., 2006); this is expected to suppress precipitation. It is considered that the balance between them determines the mean and extreme precipitation change.

The rate of increase of daily mean precipitation due to the climatology changes (Fig. 3a) was about 3% under the temperature increase of about 3.6°C at the surface in our experiments. Our result ($=3\%/3.6^\circ\text{C}\approx 0.8\%/^\circ\text{C}$) is reasonable enough in comparison with the general results of global average, that is, the rate of change of global mean precipitation with temperature is about 1–3% per 1 K [e.g. Held and Soden, 2006; IPCC AR5 WG1, chapter 7.6].

We added the description of the above physical interpretation in the revised manuscript as follows:

p.7, l.148:

“The increasing rate of mean precipitation due to the changes in climatology is about 3% under the 3.6°C warming of surface temperature from the present climate; this

value is consistent with those in previous studies (Held and Soden, 2006; Pall et al., 2007).”

p.8, l.173:

“The changes in climatology increased both the mean and extreme precipitation. The increasing rate in extreme precipitation (R1D) was larger than that in mean precipitation (RAVE); this result was consistent with those of previous studies (Emori and Brown, 2005; Pall et al., 2007). Regardless of mean and extreme precipitation, changes due to the climatology change can be explained by the thermodynamic change in the large-scale atmosphere, i.e. the precipitation increases due to the enriched water vapour in the warm atmosphere related to the Clausius-Clapeyron relationship.”

p.10, l.221:

“Most changes in the precipitation due to climatology changes can be explained by the thermodynamic change in the large-scale atmosphere associated with the Clausius-Clapeyron effect. On the other hand, the changes in climatology also include changes in the dynamic structure of the atmosphere, such as atmospheric stability. Because the temperature increase due to global warming is greater in the upper troposphere, the vertical stability increases (Bonny et al. 2006). The climatology projected in MRI-AGCM3.2 also indicates such stabilization, and the downscaled results from Pseudo-Clim-DS and Future-DDS inherited this aspect. Regarding the influence of changes in climatology on precipitation intensity, the results from the Pseudo-Clim-DS indicate that heavy precipitation was enhanced and weak precipitation was slightly suppressed. The reduction of weak precipitation seems to be attributable to the stabilization of the atmosphere in the future climate.”

3. Why do the changes in climatology not dominantly influence the mean precipitation in the target region?

In the case of our demonstration, the decrease in mean precipitation associated with the decrease in the number of cyclones surpassed the precipitation increase resulting from the balance between the Clausius-Clapeyron effect and the stabilization in warming climate. Whether climatology or perturbation changes is dominant depends on location and season as well as the size of an evaluated region.

In general, the regional climate is strongly affected by (i) how the synoptic disturbances pass through a target region and (ii) how large the evaluated region is. In the region where synoptic disturbances bring on a substantial amount of precipitation,

the impact of perturbation change on the regional precipitation strongly depends on the abovementioned two conditions. That is, the degree of influence of changes in perturbation largely varies with location and size of a target area. For example, there is a possibility that disturbances, causing most of the precipitation in the present climate in an area, hardly pass through the area in the future climate.

We added the sentences below to the revised manuscript:

p.10, l.232:

“If the changes in perturbation are considered along with those in climatology, the regional precipitation change can be determined by the balance of the thermodynamic effect, the changes in mean atmospheric structure, and changes in perturbation characteristics. Among these, the influence of perturbation changes is strongly dependent on the location and size of an evaluated area; the smaller the size of the evaluated area is, the larger the influence due to perturbation change becomes. The changes in perturbation are potentially crucial for future regional climate.”

p.10, l.245:

“In our demonstration of the proposed procedure, perturbation changes have a large impact on the precipitation change. This means that the effect of the changes in perturbation cannot be negligible for the projection in an area where the frequency and intensity of large-scale disturbances are projected to change in the future climate.”

Please note that the results in the demonstration strongly depend on the used GCM, because the projection would include uncertainty due to model bias as well as emission scenario. We added the sentences below to the manuscript:

p.11, l.260:

“In this study, only a pair of present and future climates estimated by a single GCM was used. However, there is uncertainty in future climate projections with GCMs because of imperfections regarding the emission scenario and model, amongst others. The uncertainties would be involved in both the climatology and perturbation components of future projections. By applying the procedure to multi-regional climate projections using multi-GCM projections, the procedure may also provide information on the characteristics of uncertainty in regional climate projections, such as which atmospheric component causes a spread in regional climate projections.”

4. Is it possible to apply the consequence of the demonstrated results everywhere?

From the discussion of Questions 1–3, the consequence from the demonstrated results, that is, that the perturbation change is dominant, is not always true: It depends on the target region and season. The demonstrated result strongly suggests that the changes in perturbation possibly become the main cause of the regional precipitation change. Our argument through the discussion is that our proposed method should be applied to many other regions precisely because of the dependency of the results on the target region.

We added the following in the discussion section:

p.11, l.254:

“Although many studies have concentrated on projecting the future climate and evaluating the effect of climatology changes so far, the results of our procedure emphasise the importance of the symmetric treatment of changes in climatology and perturbation along with the nonlinear effect to precisely understand the regional climate projection. By considering the influence of each change and their nonlinear effects, the procedure could provide clues to mechanisms that cause future regional climate change.”

A more detailed analysis is not within the scope of this paper, because its main purpose is the demonstration of the proposed procedure. We consider further evaluation of the physical meaning of the results from this procedure to be necessary in future studies by applying it to any other regions and seasons.

Reference:

Adachi, S. and Kimura, F. A 36-year climatology of surface cyclogenesis in East Asia using high-resolution reanalysis data. *Sci. Online Lett. Atmos. (SOLA)* **3**, 113-116 (2007). doi:10.2151/sola.2007-029.

Bony, S. *et al.* How well do we understand and evaluate climate change feedback processes? *J. Climate* **19**, 3445–3482 (2006).

Held, I.M. and Soden B.J. Robust responses of the hydrological cycle to global warming. *J. Climate* **19**, 5686–5699 (2006). doi:10.1175/JCLI3990.1.

Pall, P., Allen, M.R. and Stone, D.A. Testing the Clausius–Clapeyron constraint on changes in extreme precipitation under CO₂ warming. *Clim. Dyn.* **28**, 351 (2007). doi:10.1007/s00382-006-0180-2.

233-236: I got really confused from reading the methodology. “5-day time slice experiments”? So, these experiments are not continuous simulations? Much more detail is needed to understand this. What is the initial condition for each day’s “time slice”? How long does it take the model to equilibrate to the new initial condition, much longer than just 5 days, I think. Explain in more detail!

[Reply]

Thank you for pointing out the necessity of a detailed explanation of the time-slice experiment. Figure R2_1 shows a schematic diagram of the time-slice experiment. Each run had a 5-day integration period, which consisted of the first day for model spin-up and the last four days for analysis. For the calculation of 4-month climate, 31 runs were conducted. We used continuous analysis data from 1 June–30 September by sequentially connecting all data. We have revised the text to clarify the time-slice experiment as follows:

p.12, l.275:

“The calculation was conducted as a time-slice experiment. The calculation for four months in each year was divided into 31 runs, and each run was conducted every four days. The integration period of one run was five days, which consisted of one day for model spin-up and four days for analysis. The outputs in last four days were connected sequentially and used for analysis.”

Figure R2_1:
Schematic diagram of time-slice experiment, which explains the relationship between the integration period and analysis data.

Another concern you mentioned is whether the spin-up time is enough. Figure R2_2 indicates the composite of precipitation output during the 5-day integration for each experiment. The data were averaged in the analysis domain and all time-slice runs (31 runs x 25 years = 775 runs per experiment). There are no apparent biases in precipitation for the integration period of more than 24 h. According to this evaluation, we confirmed that a spin-up period of 1 day is enough, at least for this study.

Figure R2_2: Relationship between precipitation output and integration time. The data were averaged in the analysis domain and all time-slice runs (31 runs x 25 years = 775 runs per experiment).

Minor comments:

Title: I find the “climatology, perturbation, and their interaction” in the title (and also later in the text) a bit unprecise and perhaps even misleading. In my opinion, this is about “Contributions of changes in the mean, changes in transients, and resulting non-linearities to regional climate change”.

[Reply]

Thank you for pointing this out. We agree with your suggestion and revised the title by adding the word “changes” and modifying “their interaction” to “resulting nonlinearity”.

As you recommended, “resulting nonlinearity” may be more straightforward than “interaction”. We would defend, however, the use of “climatology” and “perturbation”, because “climatology” is widely used and understandable as a long-term mean field. The use of “perturbation” is better than that of “transient”, because the latter would induce the image of a certain state temporally moving to another. The title has therefore been revised from “Contributions of climatology, perturbation, and their interaction to regional climate change” to “Contributions of changes in climatology and perturbation and the resulting nonlinearity to regional climate change”.

Abstract: It is a bit hard to understand, in particular line 8, where “three contributions to regional climate change” are mentioned. From the previous text it is not entirely clear which three contributions are meant.

[Reply]

Thank you for this comment. We have revised the abstract to specify the “three contributions” as follows:

p.2:

“Future changes in large-scale climatology and perturbation may have different impacts on regional climate change. Because climatology and perturbation changes are mainly characterised by thermodynamic and dynamic changes, respectively, comparing the influences of these two factors is important. Although many studies have investigated the influence of climatology changes on regional climate, the significance of perturbation changes is still debatable. The nonlinear effect due to these two changes is also unknown. We propose a systematic procedure, which extracts the two effects due to changes in climatology and perturbation and the resulting nonlinear effect. We demonstrate the usefulness of the procedure, applying it to future changes in precipitation. All three effects have the same degree of influence, especially for extreme rainfall events. Thus, regional climate assessments should consider not only climatology change but also perturbation change and their nonlinearity. This procedure can advance interpretations of future regional climate.”

30: I do not entirely agree. Dynamic changes may also project on changes in the mean. For example, if the frequency or intensity of storms changes this will also project on the mean.

[Reply]

Thank you. We agree with your statement: dynamic change also affects the mean precipitation. In the same manner, thermodynamic change enables change in intensity of precipitation. To clarify this, we revised the text as follows:

p.3, l.26:

“To understand the causes of regional precipitation change, the influences of large-scale thermodynamic and dynamic changes need to be understood. Although both of these changes affect the amount and intensity of regional precipitation, few studies have compared each impact on a regional scale. We presume that the precipitation amount is mainly affected by thermodynamic changes, whereas the intensity and frequency of precipitation might rather be influenced by dynamic changes.”

40-45: See my previous comment. Also, the time scale discussion is a bit awkward.

Thermodynamic changes, which I agree are slow, can impact the rainfall intensity of a storm, which is a fast process. You say that “dynamic change corresponds to a change at a relatively short time scale”. I think this all does not really make sense here. Rainfall events are always short in nature because they are produced by storms, hence dynamics. But this is all irrelevant. I recommend removing this entire discussion about fast and slow.

[Reply]

Thank you for your comment. As you pointed out, the time-scale discussion in the previous manuscript was confusing. We removed the corresponding part. The purpose of this paragraph is to explain the relationship between thermodynamic and climatology changes, and that between dynamic and perturbation changes. We substantially revised it so as to express what we want to say in the text below:

p.3, l.39:

“Some studies have addressed these influences from another perspective (Schär et al. 1996; Kimura and Kitoh, 2007; Yoshikane et al. 2012). In these studies, a large-scale atmospheric state was decomposed into mean states and fluctuations from it. The influences of the decomposed components on regional climate were subsequently evaluated by using a regional climate model (RCM). The changes in the mean state between two climates can then be interpreted as a thermodynamic change, whereas the those in fluctuation can be seen as a dynamic change. In this study, the mean and

fluctuation components of large-scale atmosphere are called climatology and perturbation, respectively.”

45: Climatology and perturbation are not very precise expression here. I would prefer “changes in the mean ($\langle p \rangle \rightarrow \langle f \rangle$) and changes in transients ($p' \rightarrow f'$)”.

[Reply]

Thank you for this suggestion. Since the definitions of “climatology” and “perturbation” in this paper are clearly described as shown in the previous reply to your comment, we kept the use of “climatology” and “perturbation”. As described above, “climatology” is widely used as mean field at a certain period. “Perturbation” is better than “transient”, because the latter might induce the image of a temporal state moving from a certain state to another with some trend.

52-69: I find this literature review much too long and detailed. Just a few facts, that should be enough.

[Reply]

Thank you for this advice. We think that all the literature is necessary for our study. However, as you mentioned, the literature review might be too detailed and included verbose description. We revised the manuscript and eliminated redundancy.

72: Abbreviations. I think the paper use too many, too complicated, and unfamiliar abbreviations (DDS, PDDS, FDDS, PCCDS, PPCDS, etc.), which are difficult to remember. Please avoid.

[Reply]

Thank you for your suggestion. As you mentioned, too many abbreviations reduce readability for readers. On the other hand, the appropriate use of abbreviation improves the readability. To avoid the readers looking back to find the definition of experimental names, we changed them as follows:

Previous manuscript	New manuscript	Experiment description
PDDS	Present-DDS	Present climate direct downscaling
FDDS	Future-DDS	Future climate direct downscaling
PCCDS	Pseudo-Clim-DS	Pseudo climatology downscaling
PPCDS	Pseudo-Perturb-DS	Pseudo perturbation downscaling

Here, DDS is a general abbreviation for direct (dynamical) downscaling.

116-137 : I think this discussion is unnecessarily complicated. Delta-cp is simply the residual of the total response that cannot be explained by the simple linear addition of the two individual effects (Delta-C, Delta-P). Usually this is considered to be due to non-linearities of the system, and I would state it like that. The word “interaction” is misleading.

[Reply]

Thank you for your comment. According to your suggestion, we decided to use “nonlinear effect” or “nonlinearity” instead of “interaction”. As you pointed out, Δ_{cp} is regarded as the residual of the total response from the simple linear combination of the two individual effects ($\Delta C + \Delta P$). However, the physical interpretation of Δ_{cp} highlights the advantage of our method over the pseudo-warming method. In the pseudo-warming method, the total response Δ is estimated from the response to the changes in climatology only. Our method considers the response to perturbation change as well as that to the climatology change equivalently. This symmetric treatment of these two factors reveals the nonlinear effect.

In order to show the advantage in our procedure compared with the existing methods, it is important to introduce $\Delta C1$ and $\Delta P1$ (see Fig. 1 again) and clarify that the differences between $\Delta C0$ and $\Delta C1$ and between $\Delta P0$ and $\Delta P1$ are equal to Δ_{cp} . Previous studies (Yoshikane et al., 2012; Nishizawa et al., 2016) attempted to compare the influences of the climatology change with that of perturbation change directly using $\Delta C0$ and $\Delta P1$. However, this way cannot extract the accurate contribution of the perturbation change, because $\Delta P1$ includes the Δ_{cp} .

From the reason described above, this discussion part is important. However, as you pointed out, the description was somewhat complicated. We rewrote this part so as to be simpler as follows:

p.6, l.111:

“ Δ_{cp} in Eq.(4) has an important implication for physical meaning. Regarding the two large triangles parallel to the climatology axis in Fig.1, the height of the front

triangle, $\Delta C0$, is the regional climate response derived from Eq.(2). If the present perturbation is replaced by the future one under the same climatology changes, we can expect a different response, indicated as the height of the back triangle, $\Delta C1$. The difference between the two estimations equals the nonlinear effect: $\Delta cp = \Delta C1 - \Delta C0$. The same is true with regards to the perturbation change; $\Delta P0$ and $\Delta P1$, respectively, are the differences by changing the perturbation under the present and future climatology and $\Delta cp = \Delta P1 - \Delta P0$. Previous studies (Yoshikane et al. 2012; Nishizawa et al. 2016) have attempted to compare the contributions of climatology and perturbation changes by using $\Delta C0$ and $\Delta P1$. However, the nonlinear effect was not considered in these evaluations. Our procedure starts from nullifying the hypotheses of $\Delta C1 = \Delta C0$ and $\Delta P1 = \Delta P0$. The procedure considers the response to changes in perturbation as well as that to changes in climatology equivalently. This symmetric treatment of these two components reveals the nonlinear effect.”

The discussion about this procedure in the second-to-last paragraph in the text is important. This paragraph describes the relationship of sign between $\Delta C0$, $\Delta P0$, and Δcp to understand how the nonlinearity affects the expected climate, $\Delta C0 + \Delta P0$. This description is necessary for interpreting the results from the subsequent demonstration. Therefore, this description was kept.

Finally, we appreciate your comment, which we found very useful. We consequently reconsidered the discussion in this section from the reader's view. Thank you very much.

Table 2 and text: The naming of Rave is awkward. Can you put it in all capital letters? Also, in meteorology, P is usually used as variable name for rain. I recommend: P, P1D, and CDD (but not sure what it stands for).

[Reply]

According to your recommendation, we changed the word in capitals, that is, RAVE.

Figure 1: This figure is a nice idea, but it is too complex and hard to understand. Please stick to the most important aspects and remove extra information.

[Reply]

Thank you for this suggestion. All the information in Fig. 1 is important for understanding the physical meaning of the proposed method. According to your suggestion, we modified Fig. 1 to emphasise the most important aspects, for example, the relationship among Δ , ΔC_0 , ΔP_0 , and Δc_p .

223: Method. You need to explain in more detail how the BCs for the individual simulations are generated. For example, how do you calculate $\langle p \rangle$, $\langle f \rangle$, p' , and f' , and which model quantities are included. Do you use spectral nudging in your model?

[Reply]

Thank you for pointing this out. We added a more detailed explanation about the method for how the initial and boundary conditions are prepared. We did not use spectral nudging in either of the domains. Any serious detachment for large-scale dynamics between the GCM and RCM results did not occur, because the integration period of each run was relatively short (5-day integration). A detailed explanation was added to the text as follows:

p.12, l.280:

“The large-scale climate data used for the initial and boundary conditions of RCM were provided from six-hourly data in the present and future climates estimated by MRI-AGCM3.2S (Mizuta et al. 2012), explained in the next subsection. The climatology of the large-scale atmospheric state was determined via two steps. First, monthly means averaged over 25 years were prepared from May to October. Second, regarding these as the 15th day of each month, the climatology on each day was determined by their linear interpolation. Thus, the climatology varied from day to day. On the other hand, the perturbation was calculated by subtracting the obtained daily climatology from the six-hourly data. To establish the initial and boundary conditions of RCM, we combined these climatology and perturbation components for temperature, pressure, and zonal and meridional winds. The combination for each run is listed in Table 1. Specific humidity was calculated using the combined temperature under the consideration that relative humidity depends on individual perturbation. The sea surface temperature (SST), soil temperature, and soil moisture were also prepared in the same way with atmospheric boundary conditions. Note that SST was given to RCM as a constant value based on the five-day average for each simulation period. No nudging techniques were used in this simulation, because the integration period for each run was only five days.”

2nd review:

Most of my initial comments were addressed to my satisfaction. However, after re-reading the revised document, additional concerns came up. These concerns should be properly addressed before the paper can be published.

General comments:

The English language of the manuscript needs to be improved in major ways. In its current form, the paper often lacks the necessary clarity, making it hard to understand the arguments.

42-49: Here and at other locations (e.g., line 176), the manuscript states that changes in the mean would be synonymous with thermodynamic changes, and changes in the perturbations would be due to dynamic changes. I find this confusing and wrong, and I believe I made a similar comment in my first review. Changes in the dynamics of the atmosphere manifest themselves not only in terms of perturbation changes. We know there are important mean changes in the circulation, like changes in the width of the Hadley cell, the position and strength of the jets and storm tracks, or the spatial pattern of sea level pressure. All these will be reflected in climatological mean changes of u , v , Z , T , or p . Likewise, there can be temporal perturbations in temperatures or humidity, which do not affect the mean, but which may affect precipitation. So, changes in mean and changes in perturbation are not equivalent with changes in thermodynamics and changes in dynamics. I therefore strongly recommend to change the text in this respect. Later in this paragraph (line 45) the authors seem to admit to this point, but in the next sentence (line 47) they take it back again. Then, in line 48, the authors write that changes in mean precipitation are assumed to be due to thermodynamic changes. But this contradicts the findings of the paper, as it is found that changes in the intensity and frequency of dynamical systems strongly contribute to change in mean precipitation.

I still think the methodology section needs to be expanded and improved. It is important that the reader understands the details of this work. For example, it is mentioned that the experiments are “time-slice” experiments. I think the word “time-slice” is inappropriate, since it usually refers to multi-year-long simulations of the same year. Essentially, it seems that every four days five-day-long forecasts from the GCM produced initial conditions are made, using the 6-hourly GCM derived varying boundary conditions. These are not time-slice simulations. Also, geopotential is not mentioned in the list of variables in line 288-289; could you explain why? The sentence about specific humidity is unclear (line 290). Equally confusing is the next sentence about SST, soil moisture, etc., e.g., what is meant by “in the same way”. In line 293, do you mean to say that SSTs were kept constant over the course of a 5-day forecast?

Specific comments:

29: The sentence starting with “We presume ...” does not make sense. If the intensity or frequency of precipitation changes, then this clearly impacts the mean precipitation, unless

intensity and frequency changes have opposite sign and cancel each other out. I recommend removing this last sentence.

215: See comment above.

53: This sentence is not very helpful since it does not explain the “extended method” nor does it comment on its validity.

55: What is meant with “a series of the pseudo-warming method”?

64: Please explain what a “direct downscaling experiment” is.

66: Explain clearly what “component” is meant here?

71: Again, the word component is unclear. Also, the word large-scale is confusing, so maybe just remove it here.

72-74: Sentence is difficult to understand. Improve the English language.

110: What does “cp” stand for?

133: The word “simultaneously” is misleading. You make four separate simulations to extract the needed information. There is nothing simultaneous.

139: Explain the SCALE-RM is a regional climate model, and that MRO-AGCM3.2S is an atmosphere-only climate model. Else the reader won't understand what is meant.

158: The inequality is true for the area averaged quantities. It would be interesting to see how it looks like on a grid-point basis and then perhaps describe the result in words.

227: “inherited” is probably the wrong word.

255: Emphasize?

Nature communications: manuscript NCOMMS-17-12256-A

Reply to Reviewer #2's comments

Most of my initial comments were addressed to my satisfaction. However, after re-reading the revised document, additional concerns came up. These concerns should be properly addressed before the paper can be published.

General comments:

The English language of the manuscript needs to be improved in major ways. In its current form, the paper often lacks the necessary clarity, making it hard to understand the arguments.

[Reply]

Thank you for your comment. We have carefully re-read the manuscript and have also used an English language proofreading service. We attach its certificate at the end of this file. We believe that the language of the new manuscript is improved.

42-49: Here and at other locations (e.g., line 176), the manuscript states that changes in the mean would be synonymous with thermodynamic changes, and changes in the perturbations would be due to dynamic changes. I find this confusing and wrong, and I believe I made a similar comment in my first review. Changes in the dynamics of the atmosphere manifest themselves not only in terms of perturbation changes. We know there are important mean changes in the circulation, like changes in the width of the Hadley cell, the position and strength of the jets and storm tracks, or the spatial pattern of sea level pressure. All these will be reflected in climatological mean changes of u , v , Z , T , or p . Likewise, there can be temporal perturbations in temperatures or humidity, which do not affect the mean, but which may affect precipitation. So, changes in mean and changes in perturbation are not equivalent with changes in thermodynamics and changes in dynamics. I therefore strongly recommend to change the text in this respect. Later in this paragraph (line 45) the authors seem to admit to this point, but in the next sentence (line 47) they take it back again. Then, in line 48, the authors write that changes in mean precipitation are assumed to be due to thermodynamic changes. But this contradicts the findings of the paper, as it is found that changes in the intensity and frequency of dynamical systems strongly contribute to change in mean precipitation.

[Reply]

Thank you for your comments. Addressing them has provided us with an excellent opportunity to review our manuscript.

On the relationship between climatology/perturbation changes and thermodynamics/dynamics changes

We agree with your opinion that “changes in mean and changes in perturbation are not equivalent with changes in thermodynamics and changes in dynamics.” As you point out, changes in climatology and in perturbation include both thermodynamic and dynamic changes. Table R1_1 shows our reconsideration of the roles of these thermodynamic and dynamic changes in the climatology and perturbation changes from the perspective of future climate change associated with increasing greenhouse gases.

Table R1_1: Thermodynamic and dynamic changes in the climatology and perturbation changes from the perspective of future climate change associated with increasing greenhouse gases.

	Thermodynamic change	Dynamic change
Climatology change	Increase of atmospheric moisture content in association with temperature warming (Clausius-Clapeyron relationship)	Changes in long-time flow pattern associated with global circulation (e.g. width of the Hadley cell, position and strength of jets)
Perturbation change	Changes in temperature and specific humidity due to changes in track of disturbances (not include Clausius-Clapeyron effect)	Changes in frequency, intensity, and track of disturbances

First, we considered the climatology changes. The thermodynamic change affecting precipitation change corresponds to increase of atmospheric moisture content, which is associated with temperature warming, based on the Clausius-Clapeyron relationship. On the other hand, the dynamic change corresponds to changes in the large-scale pressure and flow patterns. As you point out, these include changes in meridional circulation (e.g., Hadley circulation), and in the position and strength of westerly jets.

Next, we considered the perturbation changes. The dynamic change caused by perturbation changes includes changes in the frequency, intensity, and track of disturbances (e.g., tropical and extratropical cyclones). The changes in the frequency and intensity of disturbances in a certain region result from changes in the numbers and tracks of disturbances on a global scale, while changes in the tracks of disturbances lead to the thermodynamic change of disturbances. The details of thermodynamic change caused by perturbation changes are as follows. If we assume that the track of a certain perturbation has changed under the same climatology (mean state), the perturbation would pass through different temperature and humidity environments from those of the original. For example, when the track of the perturbation is shifted to pass through more northern areas, the temperature of the mean state surrounding the perturbation would decrease. If we consider the relative humidity distribution as an attribution of disturbance, the specific humidity would decrease according to the temperature decrease. As a result, this track shift would affect the precipitation.

It is also important to note that temporal fluctuations in temperature or humidity vary by event. The differences in these fluctuations between present and future climates are mainly related to the Clausius-Clapeyron effect, which is evaluated as the thermodynamic change in the climatology component of our method. This is because we assume that the climatology component stays unchanged when we consider the thermodynamic change in the perturbation component.

Based on the above discussion, we have revised the 4th paragraph in introductory section (see Line 39-50).

Dominant factors in each of climatology and perturbation changes for precipitation change in the target region

Under the above categorization, we have reconsidered which of the thermodynamic and dynamic changes are dominant for precipitation change with regard to both climatology and perturbation changes in the target region of the demonstration run. In order to interpret the results physically, it is helpful to thus simplify the complex problem by comparing influences of each factor and discuss a priori about major factors. In this sense, we consider that at least in the mid-latitude region, thermodynamic change has greater a influence on the regional precipitation change than does dynamic change in terms of climatology, while dynamic change has a relatively larger influence in terms of perturbation change. Our reasons for this thought are explained below.

For the climatology change: Previous studies (e.g., Emori and Brown, 2005) have reported that the influence of thermodynamic change on global mean and extreme

precipitation is larger than that of dynamic change (based on the analysis of several GCM results). They concluded that dynamic change has little effect over mid- to high-latitudes. Furthermore, Figure 2 of Emori and Brown (2005) indicates that the influence of thermodynamic change on precipitation around our target region is dominant over that of dynamic change. Therefore, we conclude that thermodynamic change is a major factor affecting precipitation change due to climatology change.

For the perturbation change: As the dynamic changes in perturbation change, the changes in number of typhoons and track of extratropical cyclones are projected in future climate simulations by the GCMs in the target region (Ulbrich et al. 2008, Mizuta et al. 2012). On the other hand, we considered changes in temperature and humidity due to shifts of perturbation track as the thermodynamic change. This thermodynamic change by the perturbation change would be apparent only when the situation satisfied the following conditions simultaneously: 1) the climatology at the mean locations of storm tracks is drastically different between present and future climates, and 2) both present and future locations of storm tracks are contained within the target region. The target region in this study does not satisfy the above situations; therefore, the influence of thermodynamic change on regional precipitation seems to be smaller than that of dynamic change. On this basis, a major factor affecting regional precipitation change is the dynamic change in the perturbation change. Actually, the above is demonstrated within our results; climatology and perturbation changes are roughly regarded as the thermodynamic and dynamic changes, respectively, in the target region.

Since the contributions of thermodynamic and dynamic change in climatology and perturbation, in general, depend on which region is chosen, we think that qualitative evaluation of each factor affecting regional precipitation change is important for understanding its mechanism. However, we would like to treat this issue as a future study.

To make our argument clearer within the manuscript, we have added the following detailed explanations: 1) in terms of both climatology and perturbation changes, which of thermodynamic and dynamic changes can be thought to have relatively larger influence on regional precipitation change; 2) as a result of such comparison, the climatology and perturbation changes can be roughly viewed as the thermodynamic and dynamic changes, respectively. By this revision, we believe that there is no contradiction between our thought and finding of this study that you pointed out. The new text appears in the 5th paragraph of the Introduction section (see Line 51-58).

For more advanced of understating, precise evaluation of contributions to regional climate change from each of four factors (Table R1_1) is very important. We will address this issue in future work; however, we have highlighted its importance in our Discussion as follows:

L.247-251

“To evaluate these two effects more quantitatively, further experiments using appropriate boundary conditions are required. As an example, Kröner et al. (2016) proposed a downscaling approach to evaluate the separate contributions of vertically homogeneous warming and a vertical structure change in the temperature associated with changes in climatology to the regional climate change.”

L.280-283

“To understand the mechanisms in more detail, it would be helpful to separate the factors affecting the regional climate and evaluate their contributions quantitatively; for example, thermodynamic and dynamic changes in climatology, those in perturbation, and ground surface conditions such as land use.”

References:

- Mizuta, R. 2012: Intensification of extratropical cyclones associated with the polar jet change in the CMIP5 global warming projections, *Geophys. Res. Lett.*, 39, L19707, doi:10.1029/2012GL053032
- Ulbrich, U., J. G. Pinto, H. Kupfer, G.C. Leckebusch, T. Spangehl, and M. Reyers, 2008: Changing Northern Hemisphere Storm Tracks in an Ensemble of IPCC Climate Change Simulations. *J. Climate*, 21, 1669–1679.

I still think the methodology section needs to be expanded and improved. It is important that the reader understands the details of this work.

[Reply]

Thank you for your comment. According to your suggestion, we have revised the methodology section for experimental setup of simulations, describing in more detail the procedure. Please see the manuscript for the final form of our descriptions. Here we show the responses to your specific questions.

For example, it is mentioned that the experiments are “time-slice” experiments. I think the word “time-slice” is inappropriate, since it usually refers to multi-year-long simulations of the same year. Essentially, it seems that every four days five-day-long forecasts from the GCM

produced initial conditions are made, using the 6-hourly GCM derived varying boundary conditions. These are not time-slice simulations.

[Reply]

According to your suggestion, we decided not to use the term 'time-slice'. We removed the sentence 'The calculation was conducted as a time-slice experiment.' from the text. We believe that our explanation of the experimental procedure is sufficient without this sentence.

Also, geopotential is not mentioned in the list of variables in line 288-289; could you explain why?

[Reply]

Since MRI-AGCM is a hydrostatic model using sigma-pressure hybrid coordinates (Arakawa and Lamb 1977, Simmons and Burridge, 1981), the geopotential height is a diagnostic variable. Although the geopotential height can be calculated from pressure and temperature, we added geopotential height to the list of variables provided from MRI-AGCM, as follows:

L.306-309

"The following variables are used for the initial and boundary conditions: atmospheric temperature, geopotential height, pressure, specific humidity, zonal and meridional winds, surface pressure, skin temperature, SST, soil temperature, and soil moisture."

Also, in the MRI-AGCM description, we clarified that MRI-AGCM is the 'hydrostatic' model as follows.

L.341-342

"MRI-AGCM3.2S is a high-resolution hydrostatic atmospheric GCM developed at the Meteorological Research Institute (MRI) for climate simulations."

The sentence about specific humidity is unclear (line 290).

[Reply]

We revised our description of to create specific humidity as follows:

L.321-329

"The specific humidity is calculated from the relative humidity and the combined temperature, constructed by summing the two components. The relative humidity used for Pseudo-Clim-DS and Pseudo-Perturb-DS is assumed to depend on the climate of perturbation, namely, the relative humidity for them were obtained from present and future GCM simulations, respectively. The assumption that the relative

humidity remains unchanged between present and future climates is well-used in the pseudo-warming method (Kawase et al. 2008; Kröner et al. 2016). The assumption is based on the reports that the change in relative humidity is projected to be quite small in future warmer climates (Allen and Ingram 2002; Soden et al. 2005). The advantage of using this assumption is to be able to avoid super-saturated and super-dried conditions.”

Equally confusing is the next sentence about SST, soil moisture, etc., e.g., what is meant by “in the same way”.

[Reply]

We substantially revised our description of how to prepare the initial and boundary conditions. In the text, we first show the variables, including SST, soil temperature, and soil moisture, used for the initial and boundary conditions of the RCM, then, we explain how to construct these conditions for Pseudo-Clim-DS and Pseudo-Perturb-DS. We think that the description of our procedure has been improved. Please see the manuscript for the final form.

In line 293, do you mean to say that SSTs were kept constant over the course of a 5-day forecast?

[Reply]

Yes, SSTs were kept constant during the 5-day integration period. We have revised the description as follows.

L.309-310

“Note that the SST was given as a temporally averaged value in each integration period and then kept constant throughout a five-day simulation.”

Specific comments:

29: The sentence starting with “We presume ...” does not make sense. If the intensity or frequency of precipitation changes, then this clearly impacts the mean precipitation, unless intensity and frequency changes have opposite sign and cancel each other out. I recommend removing this last sentence.

[Reply]

Thank you for your comment. In the sentences, we state our views about which climatology change and perturbation change does have larger contribution to changes in precipitation amount and intensity. As you mentioned, total precipitation amount

(mean precipitation) is determined by the intensity of each precipitation event and frequency of each intensity event. However, if we focus on the changes in precipitation amount and intensity, the dominant factors causing these changes need not be the same between two precipitation indices. It is well known that precipitation intensity is enhanced by thermodynamic change in climatology (i.e., enriched water vapour due to the Clausius-Clapeyron relationship). Global mean precipitation amounts also increase as a result of thermodynamic effects, as shown in Emori and Brown (2005), although these changes are not determined only by moisture content in the atmosphere. On the other hand, the dynamic change in the perturbation (i.e., changes in frequency and intensity of disturbances) is directly linked to regional precipitation. Thus, we can first speculate that perturbation changes potentially have a greater contribution to precipitation change at a regional scale as compared to the global mean. This is important for understanding the regional precipitation change evaluated by the proposed procedure and when considering the assessment of future regional climate. We did not show the reason for the speculation in the previous manuscript. We have revised the text as follows:

L.59-63

“Although the changes in both the precipitation amount and intensity has been explained mainly by the changes in climatology so far, there is a possibility that changes in the perturbation also significantly influence the precipitation intensity if we focus on a limited area. This is because the changes in the track and frequency of cyclones directly affect the region. However, few studies have compared each impact on a regional scale.”

215: See comment above.

[Reply]

As explained above, this part explains the relative contributions of climatology and perturbation change to changes in mean precipitation and intensity. The result is compared with our expectation described in introduction. We decided to remain this part (Line 231-235).

53: This sentence is not very helpful since it does not explain the “extended method” nor does it comment on its validity.

[Reply]

We moved the explanation of Kröner et al. (2016) to the Discussion section because it is important for discussing about the influence of the climatology change. We deleted the original sentence from the Introduction section. The final form in the text (see Line 248-251) is described in the reply to the above comment.

55: What is meant with “a series of the pseudo-warming method”?

[Reply]

We meant the conventional pseudo-warming method and its extended method. Since the reference to the extended pseudo-warming method (Kröner et al. 2016) was removed from the Introduction, we have removed the word here.

64: Please explain what a “direct downscaling experiment” is.

[Reply]

The phrase ‘direct downscaling experiment’ is explained in the text as follows:

L.78-79

“The DDS experiments are conventional downscaling simulations, for which boundary conditions are given directly from GCM outputs.”

66: Explain clearly what “component” is meant here?

[Reply]

We use the word ‘component’ to describe climatology or perturbation. These definitions appear as ‘*In this paper, the mean and fluctuation components of the large-scale atmosphere are called climatology and perturbation, respectively*’ (see 3rd paragraph of the Introduction section). To clarify the meaning of ‘component’, we have revised this sentence as follows:

L.76-78

“... two experiments with boundary conditions exchanging either climatology or perturbation components between the two climates.”

71: Again, the word component is unclear. Also, the word large-scale is confusing, so maybe just remove it here.

[Reply]

We have revised the sentence. The components have been written in concrete terms and the word 'large-scale' has been deleted:

L.81-83

“Thus, by using this procedure, we can divide the causes of regional climate change into three factors: climatology change, perturbation change, and the nonlinear effect between them.”

72-74: Sentence is difficult to understand. Improve the English language.

[Reply]

We have improved the sentence as follows:

L.83-87

“The demonstration of the procedure shows the importance of the symmetric treatment of the changes in climatology and perturbation to precisely understand the regional climate change. At the same time, the nonlinear effect also shows some significant influence, especially for extreme rainfall events.”

110: What does “cp” stand for?

[Reply]

This is just a mathematical symbol for the nonlinear effect between climatology and perturbation changes. This is clear from the text:

L.120-123

“The nonlinear effect between the climatology and perturbation changes is not considered in the expected climate change. If the difference between the actual climate change and expected climate change is denoted as Δcp ,”

133: The word “simultaneously” is misleading. You make four separate simulations to extract the needed information. There is nothing simultaneous.

[Reply]

We have changed the word from ‘simultaneously’ to ‘at a time’. (Line 147)

139: Explain the SCALE-RM is a regional climate model, and that MRO-AGCM3.2S is an atmosphere-only climate model. Else the reader won't understand what is meant.

[Reply]

Thank you. We have modified the sentence as follows:

L.152-154

“SCALE-RM, which is an RCM, was used for the downscaling simulations. The boundary conditions for SCALE-RM were constructed from the results of MRI-AGCM3.2S, which is an atmospheric general circulation model.”

158: The inequality is true for the area averaged quantities. It would be interesting to see how it looks like on a grid-point basis and then perhaps describe the result in words.

[Reply]

The results of Δ_{cp} on a grid-point basis are shown in Figure 2g. Readers can see for themselves the spatial distribution of Δ_{cp} and its relationship with the expected future climate (Figure 2f) without a detailed description. The result on a grid-point basis is important for understanding nonlinear effects; however, we do not address this point in our manuscript because the primary purpose of this study was to propose and demonstrate our new procedure. This topic will be further analyzed in future studies.

Since the relationship between the expected future climate and Δ_{cp} differs grid by grid, we changed the sentence as follows:

L.171-174

“The relationship between the expected future climate and Δ_{cp} varies with the location. In terms of the areal mean, the nonlinear effect tends to enhance the expected climate change for RAVE because $(\Delta CO + \Delta PO) \cdot \Delta_{cp} > 0$, as shown in Fig. 3a.”

227: “inherited” is probably the wrong word.

[Reply]

We have changed the word as follows:

L.242-243

“The climatology projected in MRI-AGCM3.2 also indicates such stabilisation, and the downscaled results in Pseudo-Clim-DS and Future-DDS reflect this tendency.”

255: Emphasize?

[Reply]

We use the British spelling since Nature communications is a British publication.

Nature Research Editing Service Certification

This is to certify that the manuscript titled Contributions of changes in climatology and perturbation and the resulting nonlinearity to regional climate change was edited for English language usage, grammar, spelling and punctuation by one or more native English-speaking editors at Nature Research Editing Service. The editors focused on correcting improper language and rephrasing awkward sentences, using their scientific training to point out passages that were confusing or vague. Every effort has been made to ensure that neither the research content nor the authors' intentions were altered in any way during the editing process.

Documents receiving this certification should be English-ready for publication; however, please note that the author has the ability to accept or reject our suggestions and changes. To verify the final edited version, please visit our verification page. If you have any questions or concerns over this edited document, please contact Nature Research Editing Service at support@as.springernature.com.

Manuscript title: Contributions of changes in climatology and perturbation and the resulting nonlinearity to regional climate change

Authors: Sachiho A. Adachi, Seiya Nishizawa, Ryuji Yoshida, Tsuyoshi Yamaura, Kazuto Ando, Hisashi Yashiro, Yoshiyuki Kajikawa, and Hirofumi Tomita

Key: 2B70-C5F1-1C57-FFF1-C55C

This certificate may be verified at secure.authorservices.springernature.com/certificate/verify.

Nature Research Editing Service is a service from Springer Nature, one of the world's leading research, educational and professional publishers. We have been a reliable provider of high-quality editing since 2008.

Nature Research Editing Service comprises a network of more than 900 language editors with a range of academic backgrounds. All our language editors are native English speakers and must meet strict selection criteria. We require that each editor has completed or is completing a Masters, Ph.D. or M.D. qualification, is affiliated with a top US university or research institute, and has undergone substantial editing training. To ensure we can meet the needs of researchers in a broad range of fields, we continually recruit editors to represent growing and new disciplines.

Uploaded manuscripts are reviewed by an editor with a relevant academic background. Our senior editors also quality-assess each edited manuscript before it is returned to the author to ensure that our high standards are maintained.

Reviewer #2 (Remarks to the Author):

All my comments were addressed to my satisfaction. Therefore, I recommend to publish the paper. I only have a few very minor comments:

43: This sentence refers to thermodynamic change on top of climatology change. This is not entirely clear from how it is written. Perhaps, say: "The thermodynamic change due to the climatology change ...", or combine this sentence with the previous one.

47: ditto

322: "..., constructed by summing the two components." Not clear what this means. However, this may be a crucial point for the entire study. I understood as if RH and T are used to calculate q. If so, then there is nothing added.

326: well-used? > common

Reply to Reviewer #2's comments

All my comments were addressed to my satisfaction. Therefore, I recommend to publish the paper. I only have a few very minor comments:

[Reply]

Thank you very much for carefully reading the manuscript and giving us your comments. We revised the manuscript point by point as follows.

43: This sentence refers to thermodynamic change on top of climatology change. This is not entirely clear from how it is written. Perhaps, say: "The thermodynamic change due to the climatology change ...", or combine this sentence with the previous one.

[Reply]

Thank you for your suggestions. We revised the relevant sentence to "The thermodynamic change due to the climatology change corresponds to the increase in the atmospheric moisture content associated with temperature warming, ..."

47: ditto

[Reply]

We revised it to "The dynamic change due to the perturbation change refers to changes in the frequency, intensity, and track of disturbances such as tropical and extratropical cyclones, ..."

322: "..., constructed by summing the two components." Not clear what this means. However, this may be a crucial point for the entire study. I understood as if RH and T are used to calculate q. If so, then there is nothing added.

[Reply]

Thank you for your comments. As you point out, q is calculated from RH and T. To avoid misinterpretation, we revised the sentence as follows.

L.323

"The specific humidity is calculated from the temperature constructed by the procedure described above and the relative humidity."

326: well-used? > common

[Reply]

Thank you. We changed this as per your suggestion.